# The AMPK-related kinase NUAK1 controls cortical axons branching by locally modulating mitochondrial metabolic functions

Marine Lanfranchi[1,8], Sozerko Yandiev[1,8], Géraldine Meyer-Dilhet[1], Salma Ellouze[1,2], Martijn Kerkhofs[1], Raphael Dos Reis[1], Audrey Garcia[1], Camille Blondet[1], Alizée Amar[1], Anita Kneppers[1], Hélène Polvèche[3,4], Damien Plassard[5], Marc Foretz[6], Benoit Viollet[6], Kei Sakamoto[7], Rémi Mounier[1], Cyril F. Bourgeois[3], Olivier Raineteau[2], Evelyne Goillot[1] & Julien Courchet[1] ✉

The cellular mechanisms underlying axonal morphogenesis are essential to the formation of functional neuronal networks. We previously identified the autism-linked kinase NUAK1 as a central regulator of axon branching through the control of mitochondria trafficking. However, (1) the relationship between mitochondrial position, function and axon branching and (2) the downstream effectors whereby NUAK1 regulates axon branching remain unknown. Here, we report that mitochondria recruitment to synaptic boutons supports collateral branches stabilization rather than formation in mouse cortical neurons. NUAK1 deficiency significantly impairs mitochondrial metabolism and axonal ATP concentration, and upregulation of mitochondrial function is sufficient to rescue axonal branching in NUAK1 null neurons in vitro and in vivo. Finally, we found that NUAK1 regulates axon branching through the mitochondria-targeted microprotein BRAWNIN. Our results demonstrate that NUAK1 exerts a dual function during axon branching through its ability to control mitochondrial distribution and metabolic activity.

Axon morphogenesis is a multistep process culminating in the formation of axonal branches to establish a network, later refined through a process of selection of functional contacts and elimination by pruning[1,2]. The formation and stabilization of axonal branches, collectively referred to as 'axon branching', relies on the activation of intracellular signaling pathways mediating a cascade of cellular events —such as cytoskeleton remodeling, addition of membrane, and local protein translation[3–6]—that are highly taxing energetically and are thought to induce a local increase in the metabolic turnover. Evidence suggests that the rapid consumption of metabolic molecules such as

[1]Univ Lyon, Université Claude Bernard Lyon 1, CNRS, Inserm, Physiopathologie et Génétique du Neurone et du Muscle, UMR5261, U1315, Institut Neuro-MyoGène, 69008 Lyon, France. [2]Univ Lyon, Université Claude Bernard Lyon 1, Inserm, Stem Cell and Brain Research Institute U1208, 69500 Bron, France. [3]Laboratoire de Biologie et Modelisation de la Cellule, Ecole Normale Superieure de Lyon, CNRS, UMR 5239, Inserm, U1293, Universite Claude Bernard Lyon 1, 46 allée d'Italie F-69364, Lyon, France. [4]CECS/AFM, I-STEM, 28 rue Henri Desbrüères, F-91100 Corbeil-Essonnes, France. [5]CNRS UMR 7104, INSERM U1258, GenomEast Platform, Institut de Génétique et de Biologie Moléculaire et Cellulaire (IGBMC), Université de Strasbourg, Illkirch, France. [6]Université Paris Cité, CNRS, Inserm, Institut Cochin, Paris, France. [7]Novo Nordisk Foundation Center for Basic Metabolic Research, University of Copenhagen, Copenhagen 2200, Denmark. [8]These authors contributed equally: Marine Lanfranchi, Sozerko Yandiev. ✉e-mail: julien.courchet@univ-lyon1.fr

ATP limits their spatial diffusion in the axon[7,8], implying that cellular energy production must match the local demand. Understanding how these local mechanisms are set up and how they contribute to aspects of neuronal development such as the shaping of axonal complexity is therefore a key question in developmental neuroscience.

The AMP-activated protein kinase (AMPK) is a major sensor of cellular energy status, which is activated upon various metabolic stresses that lower the pool of ATP. Recently, AMPK was linked to the control of mitochondrial anchoring at presynaptic boutons in mature cortical neurons[9], coupling local metabolic needs with mitochondrial positioning. Yet, AMPK activity is largely dispensable for cortical development[10], suggesting that other metabolic regulators are involved in early axonal development. We previously identified that a signaling pathway formed by the kinases LKB1/STK11 and the AMPK-related kinase (AMPK-RK) NUAK1/ARK5 controls cortical axon branching through the regulation of mitochondrial capture at immature presynaptic sites[11]. Mutations of *NUAK1* are associated with several neurodevelopmental disorders including Autism Spectrum Disorders (ASD)[12,13], Attention Deficit/Hyperactivity Disorders (AD/HD)[14], cognitive impairment[15], and hydrocephaly[16]. In mice, *Nuak1* heterozygosity affects cortical development, leading to an array of anatomical and behavioral deficits[17]. Although the cellular functions of NUAK1 are not fully understood, it is linked to the control of the Protein Phosphatase 1 activity through the regulation of regulatory subunits, such as MYPT1[18] to regulate actomyosin contractility, or PNUTS[19], linking NUAK1 to spliceosome assembly and RNA elongation. In parallel, c-Myc over-expressing cancer cells rely on *NUAK1* expression to maintain metabolic homeostasis and for survival[20,21], suggesting a metabolic role for NUAK1. Yet the mechanism by which NUAK1 regulates mitochondrial activity and its relevance for neuronal development remain unknown.

In the present study, we determine a novel role for NUAK1 in the regulation of mitochondrial metabolism in the developing cortex. Through time-lapse imaging and Chromophore-Assisted Light Inactivation (CALI) of mitochondria[22,23], we observed that mitochondria are required for branch stabilization rather than branch initiation in cortical pyramidal neurons (PNs). We furthermore report that NUAK1 controls not only mitochondrial trafficking, but also mitochondrial metabolic activity. Using pharmacological and genetic approaches, we provide causal evidence between mitochondrial activity and NUAK1-dependent axon branching in vitro and in vivo. Finally, we report a NUAK1-associated transcriptional signature and identify that NUAK1 controls mitochondrial activity and axon branching through the regulation of BRAWNIN, a mitochondria-targeted 'microprotein' involved in the formation and stabilization of the respiratory chain supercomplexes[24,25]. Overall, our results demonstrate that NUAK1 exerts a dual effect on mitochondrial trafficking and activity in cortical PNs through its previously unknown effector BRAWNIN, and provide a novel two-hit model by which branch stabilization relies not only on the proper localization of mitochondria, but also on their local activity.

## Results

### Mitochondrial recruitment to synaptic boutons supports branch stabilization rather than branch formation

In cortical PNs, axon branching depends upon the recruitment of mitochondria to nascent presynaptic boutons[11]. However, the position of boutons-associated mitochondria relative to axon branchpoints, and the dynamics of recruitment of mitochondria relative to branch formation, has not been characterized. To test this, we first expressed mitochondria-targeted DsRed and the synaptic vesicle marker vGlut1-GFP as a marker of presynaptic boutons, then performed an extensive correlation of mitochondrial position and presynaptic boutons along the axon after 5 days in vitro (DIV) (Fig. S1A–C). As previously demonstrated, virtually all branches had a vGlut1-positive cluster at their origin, and more than half of them also showed resident mitochondria (Fig. S1D–G). More interesting is the dynamic of these

associations, which we investigated by performing time-lapse imaging of mitochondria over 24-h periods (Fig. 1A, B). Immediately following the turning of filopodia to new axonal branches (as defined by the formation of a growth-cone like structure, $T = 0$ min), mitochondria started clustering within 5 μm of branch origin (Fig. 1B and Supplementary Movie 1). Mitochondrial dwell time at branch origin was significantly longer than at other positions along the axon (Fig. 1C). Furthermore, mitochondria entered newly formed collateral branches shortly after the initial clustering at branch origin (Fig. 1B, $T = 50$ min and Fig. S1H–J). On average, there were more mitochondria in branches during growth period than during retraction (Fig. 1D), and the entry of mitochondria into branches correlated with increased length and lifetime (Fig. 1E, F). On the contrary fewer mitochondria relocated to the origin of transient branches, and remained there for a shorter time (Fig. S1I, J).

Our results demonstrate that mitochondria follow rather than precede the formation of collateral branches, which suggests that mitochondria are involved in axonal branch stabilization rather than branch formation in mouse cortical PNs. To test this, we performed Chromophore-Assisted Light Inactivation (CALI)[22] to assess the consequences of local mitochondria removal on axon branches. In HeLa cells, the expression of the genetically-encoded photosensitizer Killer-Red (mito-KR) targeted to the mitochondrial matrix led to a rapid (< 30 min after illumination) production of Reactive Oxygen Species (ROS) in mitochondria (Fig. S2A–D), accompanied by their fragmentation, specifically forming ring-shaped structures (Fig. S2E–G). ROS accumulation and mitochondrial fragmentation was restricted to the Region Of Interest (ROI) (Fig. S2B–E). Furthermore, there was no ROS increase in the cytosol, suggesting minimal ROS leakage from damaged mitochondria (Fig. S2H, I). CALI on control fluorescent protein mt-DsRed did not induce significant morphological changes in mitochondria or accumulation of ROS (Fig. S2D–G).

We subsequently expressed mito-KR in cortical PNs and targeted axonal mitochondria for photosensitization. As observed in HeLa cells, a 30 s laser pulse induced a rapid accumulation of ROS in mitochondria, restricted to the ROI (Fig. 1G, H). To assess local metabolic consequences of axonal CALI, we used PercevalHR[26], a genetically-encoded biosensor for the ATP:ADP ratio. Following laser illumination, we measured an immediate drop in the ATP:ADP ratio (Fig. 1I), which lasted for at least 60 min after illumination and was largely restricted to the ROI (Fig. 1J). In contrast, PercevalHR signal was not affected by laser illumination upon expression of control mt-DsRed (Fig. 1K). Finally, we observed branch dynamics in axons over 12 h time-lapse movies following CALI (Fig. 1L). Localized photoinactivation of mitochondria significantly altered branch dynamics in the ROI, by promoting branch retraction and decreasing branch growth, whereas this effect was not observed on the same axon outside of the ROI, indicating that overall axonal viability is not affected (Fig. 1M–O). In contrast, laser illumination had little effect on branch dynamics upon expression of mt-DsRed (Fig. 1N–P). Taken together, our results demonstrate that axonal mitochondria are required locally to support branch stabilization, presumably through metabolic regulation.

### Collateral branching, but not axon elongation, relies on mitochondrial metabolism

We next sought to identify how a modulation of glucose metabolism in cultured neurons can affect axon morphogenesis. There was a marked dose-response effect of medium glucose concentration on axon development (Fig. 2A, B). Specifically, axonal length increased as a function of glucose concentration (Fig. 2C). In contrast, even low concentrations of glucose were able to sustain a high number of collateral branches (Fig. 2D). Even in the absence of glucose, directly fueling the tricarboxylic acid cycle with a low dose of pyruvate (1 mM) was sufficient to support axonal branching (Fig. 2D). Importantly, high glucose concentration such as the ones classically used for primary

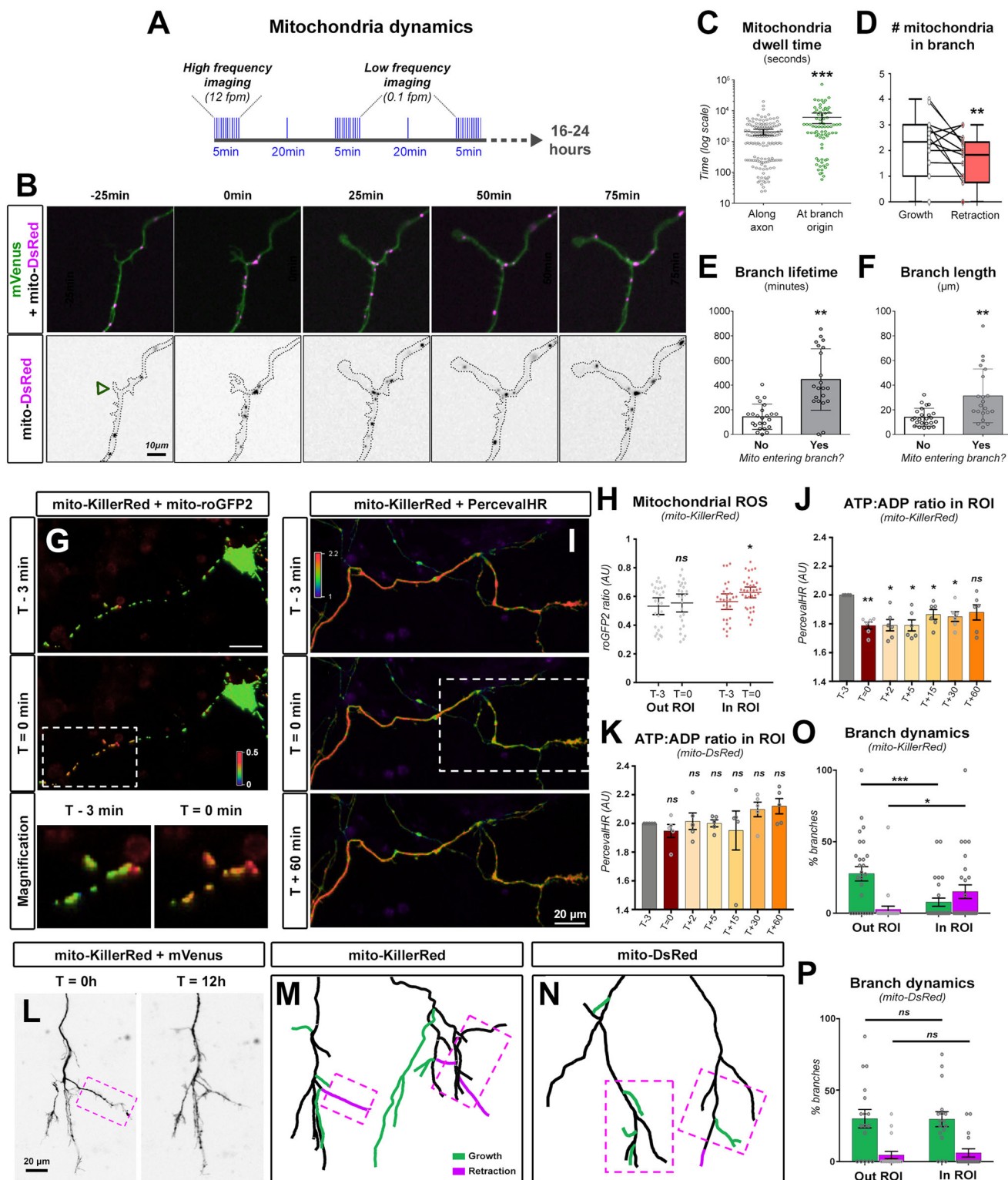

cortical and hippocampal neuronal culture favors the production of cellular ATP through glycolysis, whereas mitochondrial metabolism becomes more substantial in lower glucose concentrations that are closer to physiological conditions[27]. Combined with our results, this suggests that culture conditions that favor mitochondrial metabolism are sufficient to support axon branching.

To test this hypothesis further, we used low doses of rotenone, an inhibitor of OXPHOS complex I, to downregulate mitochondrial oxidative phosphorylation capacity without affecting glycolysis. This led to a decrease of both axon length and collateral branching (Fig. 2E−G).

Conversely, we bypassed glycolysis either by replacing glucose by galactose (a 6-carbon sugar catabolized by the glycolytic machinery that negates the net ATP production of glycolysis), or by adding pyruvate to a no-glucose medium. In both case, axon branching was largely equivalent to control conditions (25 mM glucose) despite a strong adverse effect on axon length (Fig. 2H, I). Taken together, our results show distinct metabolic requirements for axon elongation and collateral branching and suggest that mitochondrial oxidative metabolism is necessary and sufficient to support axon branching in cortical PNs.

**Fig. 1 | Presynaptic sites and mitochondria clustering at axonal branchpoints.**
**A** Strategy for long-time imaging of axonal mitochondria. Fpm: frame *per* minute.
**B** Spontaneous branch formation from the axon of a cortical PN. Branch apparition defined by length > 5 µm and a growth-cone-like structure was considered T = 0 min. Arrowhead: site of branch formation. **C** Mitochondria dwell-time at branch origin (±5 µm) compared to dwell-time along the axon. $N_{(along\ axon)}$= 143, $N_{(branch\ origin)}$= 75 out of 14 axons. Data: median ± 95% CI. Statistical test: two-tailed Mann−Whitney. **D** Mitochondria occupancy in a branch during periods of growth or retraction. Dots represent average over period (per branch), *N* = 15 branches. Statistical test: two-sided Wilcoxon matched-pairs ranked test. Data: box-and-whisker plots. The horizontal line inside the box is the median value. The endpoints of the box represent the 25th−75th percentiles, with whiskers at minimum and maximum values. Correlation of mitochondria entry into a branch with lifetime (**E**) and length (**F**). Data: median ± 95% CI. Each dot represents a spontaneous branch formation event. $N_{(No)}$= 24; $N_{(Yes)}$= 21. Statistical test: two-tailed Mann−Whitney. **G** Typical neuron expressing mito-KillerRed. CALI was performed at T = 0 min in the

Region of Interest (ROI, white box). Reactive Oxygen Species (ROS) were measured by mito-roGFP fluorescence (red = oxidized, blue = reduced). **H** Mitochondria ROS accumulation outside and inside of the ROI before and after CALI. Data: average ±95% CI. Analysis: unpaired two-tailed *t* test. $N_{(T\text{-}3,\ OUT)}$ = 26, $N_{(T=0,\ OUT)}$ = 25, $N_{(T\text{-}3,\ IN)}$ = 27, $N_{(T=0,\ IN)}$ = 34 mitochondria out of 7 independent experiments.
**I** Measurement of the ATP:ADP ratio with PercevalHR in the axon. **J, K** Average PercevalHR signal in the ROI. Data: average ± SEM. Analysis: repeated measures one-way ANOVA with Dunnett's multiple comparison (two sided). *N* = 6 axons (**J**) or 5 axons (**K**) out of 5 independent experiments. **L** Time-lapse imaging of axon growth over 12 h (magenta box). **M, N** Examples of axon (schematic reconstruction). Growing/new branches (green) and retracting branches (magenta) were quantified over 12 h (**O, P**). Data: average ± SEM. Analysis: two-way ANOVA with Bonferroni's multiple comparison test. $N_{(KillerRed)}$ = 27 axons, $N_{(DsRed)}$ = 17 axons out of 4 independent experiments. ns not significant, *$p \leq 0.05$, **$p \leq 0.01$, ***$p \leq 0.001$. Source data are provided as a Source Data file.

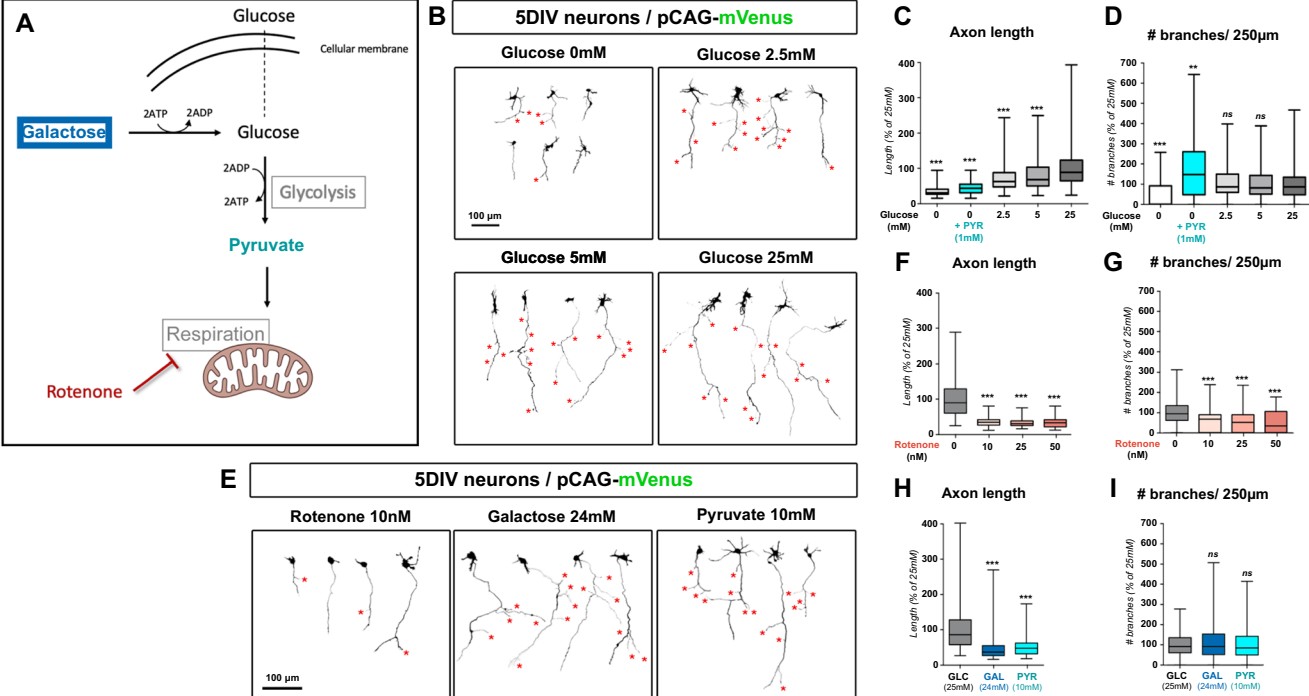

**Fig. 2 | Differential requirement of glycolysis and oxidative metabolism for axon growth and branching. A** Main pathways of glucose metabolism. Galactose was used to negate the metabolic effect of glycolysis. Pyruvate was used to support mitochondrial metabolism in a glucose-free medium. Low doses of rotenone were used to lower mitochondrial Complex I activity. Created with BioRender.com.
**B** Representative images of mVenus-expressing cortical neurons (5DIV) cultured in neurobasal medium with the indicated concentrations of glucose. Red stars: collateral branches. **C−I** Quantification of axon length and collateral branches of 5DIV neurons in the indicated conditions: **C, D** medium containing increasing concentrations of glucose or pyruvate (PYR). **F, G** 25 mM glucose-containing medium

with increasing doses of rotenone. **H, I** medium containing either glucose (25 mM, i.e., GLC), galactose (24 mM) + glucose (1 mM, i.e., GAL), or pyruvate (10 mM) without glucose (i.e., PYR). Data: box-and-whisker plots. The horizontal line inside the box is the median value. The endpoints of the box represent the 25th−75th percentiles, with whiskers at minimum and maximum values. Statistical test: Kruskal-Wallis test with Dunn's post-test (each condition compared to untreated 25 mM condition). **C, D** $N_{(0mM)}$= 48, $N_{(0mM+Pyr)}$= 75, $N_{(2.5mM)}$= 244, $N_{(5mM)}$= 191, $N_{(25mM)}$= 238. **E, F** $N_{(0nM)}$= 120, $N_{(10nM)}$= 102, $N_{(20nM)}$= 82, $N_{(50nM)}$= 44.
**G, H** $N_{(Glucose)}$= 246, $N_{(Galactose)}$= 248, $N_{(Pyruvate)}$= 130. ns not significant, *$p \leq 0.05$, **$p \leq 0.01$, ***$p \leq 0.001$. Source data are provided as a Source Data file.

## NUAK1 kinase controls axonal mitochondrial metabolism

We previously identified that the kinase NUAK1 regulates axon branching in cortical PNs through the control of mitochondrial capture at presynaptic boutons[11]. In light of this, we aimed at testing a potential role of NUAK1 in controlling mitochondrial metabolism, as recently shown in cancer models[20,21]. Through microplate-based respirometry, we measured the glycolytic and respiratory capacity of *Nuak1*[+/+] (WT), *Nuak1*[+/-] (HET) or *Nuak1*[-/-] neurons (KO). Whereas glycolysis was largely unaffected, our results revealed a trend toward a lower basal respiratory rate, as well as a significant decrease in the maximal respiratory rate upon deletion of *Nuak1* (Fig. 3A−D).

Importantly there was no difference in mitochondrial DNA content in *Nuak1* KO neuronal cultures as compared to WT conditions (Fig. 3E). In parallel, we detected a reduction in the activity of the Citrate Synthase (CS), a key enzyme of the tricarboxylic acid (TCA) cycle, from cortices of *Nuak1* KO embryos (Fig. 3F), thus confirming that *Nuak1* deletion also alters mitochondrial function in vivo.

To further confirm that *Nuak1* inactivation disrupts neuronal mitochondrial metabolic activity, we used conditional KO (cKO) mice by breeding with the glutamatergic neuron-specific Nex[CRE] line[28]. We observed a similar reduction in basal (albeit not statistically significant) and maximal (significant) respiration, further demonstrating that

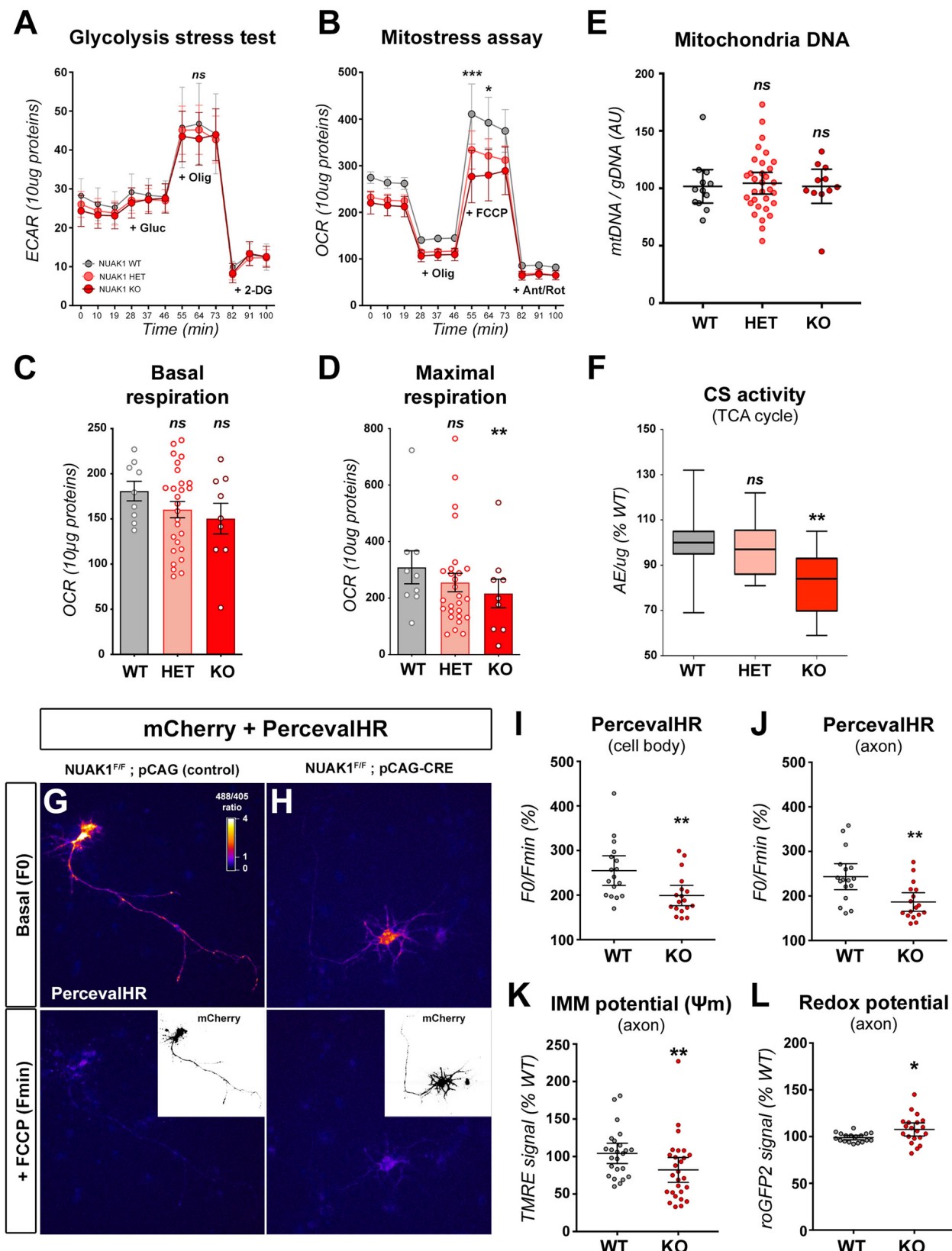

neuronal respiration is affected by *Nuak1* deletion (Fig. S3A–C). Total mitochondrial content (measured by TOM20) was unchanged, nor was the expression of proteins involved in mitochondria biogenesis (PGC1α), or fusion/fission (DRP1, MFN1, MFN2, OPA1) (Fig. S3D, E). Furthermore, we did not detect changes in markers of autophagy, suggesting that mitochondrial turnover was not altered (Fig. S3D, E).

*Nuak1* inhibition had no effect on neuronal soma size, nor on mitochondria density in the soma and axon (Fig. S3F–H).

The decreased respiratory rate suggests that NUAK1 is required not only for mitochondrial trafficking, but also for proper mitochondrial function in axons. To test this, we turned to microscopy-based strategies to directly investigate neuronal mitochondrial metabolism

**Fig. 3 | Mitochondrial metabolism is altered in NUAK1-deficient neurons.**
**A** ExtraCellular Acidification Rate (ECAR) normalized to protein dosage in 7DIV cortical neurons. Statistical test: Kruskal-Wallis test with Dunn's post-test. Data: average ± SEM. $N_{(WT)}$ = 5, $N_{(HET)}$ = 16, $N_{(KO)}$ = 5. **B** Oxygen Consumption Rate (OCR) normalized to protein dosage in 7DIV cortical neurons. Statistical test: two-way ANOVA with Bonferroni's multiple comparison. Data: average ± SEM. **C, D** Basal and Maximal respiration were extracted from (**B**). Statistical test: two-way ANOVA with Bonferroni's multiple comparison. Data: average ± SEM. $N_{(WT)}$ = 9, $N_{(HET)}$ = 27, $N_{(KO)}$ = 9. Glucose (Gluc), Oligomycin (Olig), 2-Deoxy-D-Glucose (2-DG), Carbonyl cyanide p-trifluoromethoxyphenyl-hydrazone (FCCP), Antimycin A/Rotenone (Ant/Rot). **E** Measurement by qPCR of mitochondrial and genomic DNA in cortical neurons cultures. Data: average ± 95% CI. Analysis: Kruskal-Wallis test with Dunn's post-test. $N_{(WT)}$ = 12, $N_{(HET)}$ = 32, $N_{(KO)}$ = 11 distinct cultures. **F** Citrate Synthase enzymatic activity relative to protein quantity (normalized to WT). Analysis: one-way ANOVA with Dunnett's multiple comparison test. $N_{(WT)}$ = 13, $N_{(HET)}$ = 8, $N_{(KO)}$ = 10 independent samples. Data: box-and-whisker plots. The horizontal line inside the box is the median value. The endpoints of the box represent the 25th−75th percentiles, with whiskers at minimum and maximum values. ATP:ADP ratio in $Nuak1^{F/F}$ neurons transfected with a control (**G**) or a CRE coding plasmid (**H**). Excerpt: mCherry fluorescence was used to determine neuronal morphophogy. PercevalHR signal is displayed as a fluorescence ratio between 488 and 405 illumination using a Fire LUT (White: high ratio. Blue: low ratio). **I, J** PercevalHR signal (F0) normalized by the minimal value after 5 min treatment with FCCP (Fmin). Analysis: unpaired *t* test. Data: average ± 95% CI. $N_{(WT)}$ = 17, $N_{(KO)}$ = 17 neurons out of 2 independent experiments. Raw data are provided in Supplementary Fig. 11. **K, L** Mitochondrial membrane potential ($\Delta\Psi_m$) and redox potential in the axon of $Nuak1^{F/F}$ neurons expressing a control (WT) or CRE-expressing plasmid (cKO). Data: average ± 95% CI. Analysis: unpaired two-tailed *t* test with Welch's correction. **K** $N_{(WT)}$ = 24, $N_{(KO)}$ = 27 neurons out of 5 independent experiments. **L** $N_{(WT)}$ = 18, $N_{(KO)}$ = 20 neurons out of 4 independent experiments. ns not significant, *$p \leq 0.05$, **$p \leq 0.01$, ***$p \leq 0.001$. Source data are provided as a Source Data file.

with a subcellular resolution. First, we assessed the overall metabolic consequence of *Nuak1* knockout with PercevalHR, a biosensor for the cellular ATP:ADP ratio. The rapid decrease of PercevalHR signal upon application of Carbonyl cyanide-p-trifluoromethoxyphenylhydrazone (FCCP), a potent uncoupler of mitochondrial oxidative phosphorylation, indicates that the ATP:ADP ratio reflects mitochondrial oxidative metabolism (Fig. 3G, H). By comparing the basal (F0) and minimum (Fmin, following FCCP application) fluorescence, we observed a significant decrease of PercevalHR signal in the soma (−21,9%) and in the axon (−23.4%) of $Nuak1^{F/F}$ neurons electroporated with CRE (*Nuak1* cKO neurons) (Fig. 3I, J), indicating a reduction of the ATP:ADP ratio in absence of NUAK1. Next using Tetra-Methyl-Rhodamine Ethyl-ester (TMRE), we measured a marked decrease in mitochondrial inner membrane potential ($\Delta\Psi_m$) in the axon of *Nuak1* cKO neurons (Fig. 3K and Fig. S4A). In parallel, we expressed a mitochondria-targeted roGFP2 and observed an increased ROS production by axonal mitochondria in *Nuak1* cKO neurons (Fig. 3L and Fig. S4B). Importantly, there was no difference in either TMRE and roGFP2 fluorescence in wild-type neuronal cultures when we sorted out mobile versus stationary mitochondria (Fig. S4C–F). From this result, we conclude it is highly unlikely that the low $\Delta\Psi_m$ and oxidized mitochondria state is a secondary consequence of increased trafficking in *Nuak1* cKO neurons. Altogether, our results demonstrate that NUAK1 regulates axonal mitochondrial metabolism.

### Differential roles of AMPK and NUAK1 on mitochondria association to presynaptic boutons and axon branching

We next investigated if our observations were specific to NUAK1 or shared by other AMPK-RKs. Specifically, we tested the role of AMPK, which was recently linked to the control of mitochondrial motility[29] and association of mitochondria with synaptic boutons in mature (synaptically active) cortical neurons (14DIV)[9]. Following CRE expression in $Ampk\alpha1^{F/F}$; $Ampk\alpha2^{F/F}$ (hereafter *Ampk* cKO) cortical PNs, we observed longer axons (Fig. S5A, B) and no effect on axon branching (Fig. S5C). We next performed In Utero Cortical Electroporations (IUCE)[30] of mVenus-coding plasmids in embryos with a strongly reduced AMPK activity (Fig. S5D–G). After CRE electroporation, there was no gross defect in neuronal migration or polarization (Fig. S5D, E). In addition, axonal branching was normal, both on the ipsilateral and contralateral sides (Fig. S5D–G). Thus, our results rule out a role for AMPK in supporting axon branching in cortical PNs.

We next tested whether AMPK activators promote mitochondria association with nascent presynaptic boutons in immature neurons (5DIV), as reported in mature neurons[9]. The treatment with the AMP analog 5-AminoImidazole-4-CarboxAmide Ribonucleotide (AICAR) resulted in a modest, but significant, increase in the association of mitochondria with vGlut1-positive puncta (Fig. 4A, B). Both short (90 min) and long (72 h) exposure to AICAR increased the association of mitochondria to presynaptic boutons (Fig. 4D). On the contrary, treatment with potent and direct allosteric AMPK activators A-796662 and Compound 991 (c-991) (that bind to the allosteric drug and metabolism site, a deep cleft between the β-carbohydrate-binding module and the N-lobe of the α-kinase domain)[31] had no significant effect on the distribution of axonal mitochondria (Fig. 4C–E). AICAR is a pro-drug which is converted intracellularly to the AMP-mimetic ZMP and has been reported to elicit a number of AMPK-independent actions[32,33]. Accordingly, it has been suggested that AICAR could increase the activity of NUAK1 in cancer cells[34]. In full support to this observation, there was no change in the distribution of mitochondria in *Nuak1* cKO neurons after treatment with AICAR (Fig. 4F). In sharp contrast, the effect of AICAR was retained in *Ampk* cKO neurons (Fig. 4F). To confirm that AICAR activates NUAK1, we co-transfected the LKB1-deficient HeLa cells[35] with plasmids encoding NUAK1 and its upstream activator LKB1, then treated the cells with AICAR or c-991. AICAR treatment led to a robust increase not only in AMPK, but also in NUAK1 phosphorylation at the Thr211 site, whereas c-991 only had an effect on AMPK but not on NUAK1 (Fig. 4G, H). Collectively, our results demonstrate that AICAR is a potent activator of NUAK1 and support the conclusion that its effects on axons are due to activation of NUAK1, but not AMPK.

### Upregulation of mitochondrial function rescues axon branching in NUAK1 deficient neurons

We next devised rescue strategies to upregulate mitochondrial function in *Nuak1* cKO neurons. Based on published studies, we treated cortical PNs with L-Carnitine (L-Car)[36,37], a cell-permeant, lysine-derived quaternary amine involved in the metabolism of long-chain fatty acids by the mitochondria. As previously reported[11], NUAK1-null neurons had shorter axons with fewer collateral branches (red stars in Fig. 5A). L-Car had no effect on axon length in control or NUAK1-null cortical PNs (Fig. 5B). In contrast, when normalized to axonal length, L-Car induced a complete rescue of collateral branches in NUAK1 deficient neurons (Fig. 5B, C). As an alternative mean to promote mitochondrial function, we used the mitochondria-targeted electron carrier vitamin K2/menaquinone[38]. Similar to the effect of L-Car, we observed a normalization of axonal branching without affecting axonal length in neurons treated with menaquinone-4 (MK4), an active form of vitamin K2, but not with the inactive form menaquinone-3 (MK3) (Fig. S6A, B).

We next measured TMRE uptake by mitochondria in 5DIV neuronal cultures. L-Car treatment normalized $\Delta\Psi_m$ in *Nuak1* KO neurons (Fig. 5D), indicating a rescue of mitochondrial function. Interestingly, L-Car failed to rescue the increased mito-roGFP2 signal in *Nuak1*-null neurons (Fig. 5E), ruling out that L-Car promotes axonal branching through ROS scavenging. Accordingly, the ROS scavenger N-AcetylCystein (N-AC) did not rescue axon branching in *Nuak1*-deficient neurons despite the normalization of oxidative stress in treated

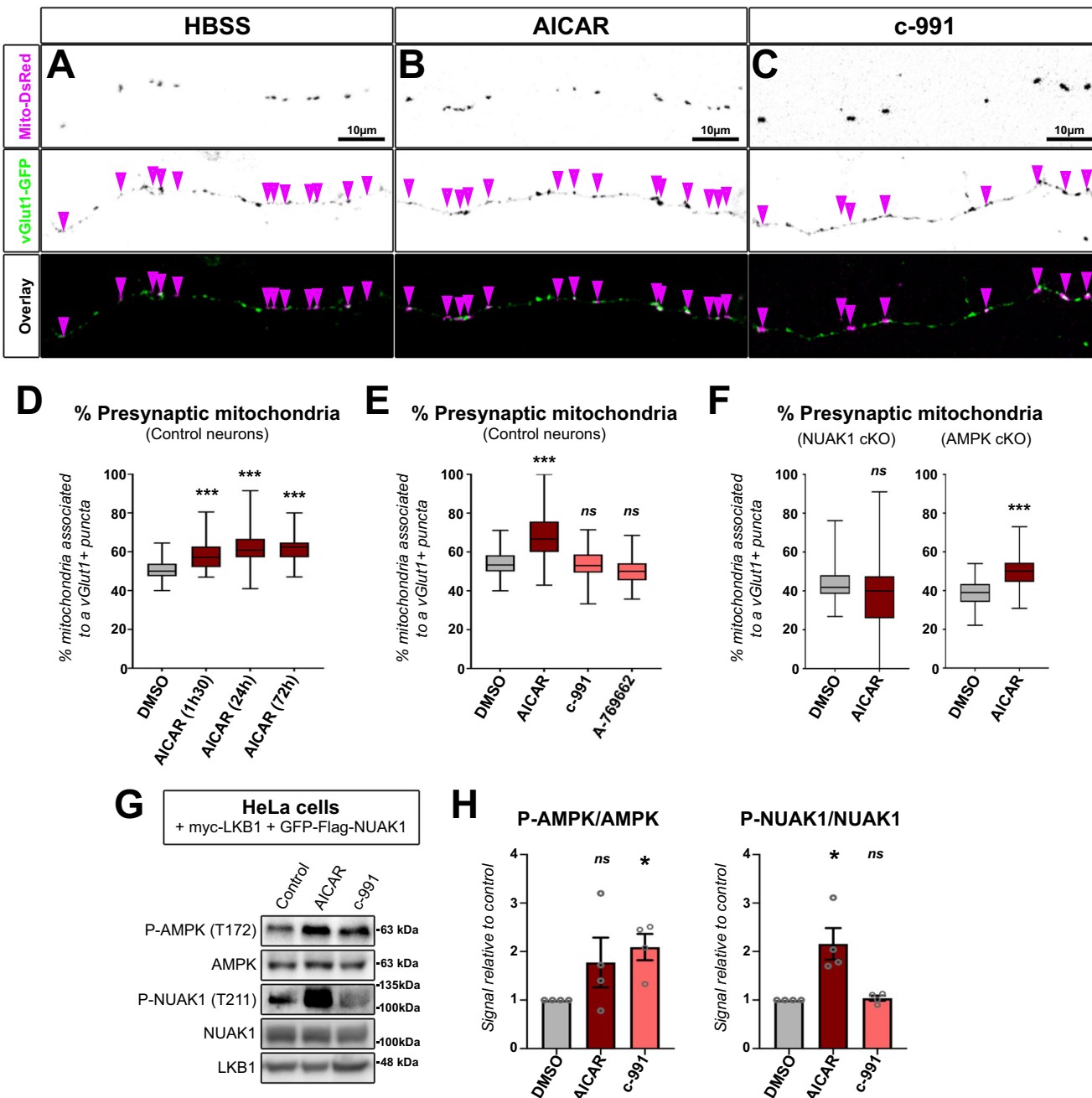

**Fig. 4 | AICAR mediates mitochondria redistribution to presynaptic boutons through activation of NUAK1 in immature cortical neurons.** Representative images of axon segments from 5DIV neurons in control conditions (**A**) or following treatment with AICAR (1 mM) (**B**) or c-991 (10 μM) (**C**). Arrowheads indicate the position of mitochondria. **D** Quantification of axonal mitochondria association to presynaptic bouton in neurons treated with AICAR for the indicated amount of time. Analysis: one-way ANOVA with Dunnett's multiple comparison test. $N_{(DMSO)} = 140$, $N_{(1h30)} = 73$, $N_{(24h)} = 63$, $N_{(72h)} = 20$ neurons from 3 independent experiments. Data: box-and-whisker plots. The horizontal line inside the box is the median value. The endpoints of the box represent the 25th–75th percentiles, with whiskers at minimum and maximum values. **E** Mitochondria association to pre-synaptic bouton after 72 h exposure to the indicated drugs. Analysis: one-way ANOVA with Dunnett's multiple comparison test. $N_{(DMSO)} = 335$, $N_{(AICAR)} = 150$, $N_{(c-991)} = 168$, $N_{(A-769662)} = 226$ individual neurons from 2 (A-769662) to 4 (other conditions) independent experiments. Data: box-and-whisker plots. The horizontal line inside the box is the median value. The endpoints of the box represent the 25th

–75th percentiles, with whiskers at minimum and maximum values. **F** Mitochondria association to presynaptic bouton in *Nuak1*[F/F] or *Ampkα1*[F/F]; *Ampkα2*[F/F] neurons transfected with a CRE-coding plasmid and following 72 h exposure to the indicated drugs. Analysis: one-way ANOVA with Dunnett's multiple comparison test. $N_{(DMSO)} = 103$, $N_{(AICAR)} = 48$ for Nuak1 cKO, $N_{(DMSO)} = 116$, $N_{(AICAR)} = 49$ for *Ampk* cKO from 3 independent experiments. Data: box-and-whisker plots. The horizontal line inside the box is the median value. The endpoints of the box represent the 25th–75th percentiles, with whiskers at minimum and maximum values. **G** Western blot of protein extracts from HeLa cells co-transfected with NUAK1 and LKB1, and treated with AICAR (1 mM) or c-991 (10 μM) for 90 min before cell lysis. **H** Quantification of 4 independent experiments. Original (uncropped) blots are provided in Supplementary Fig. 10. Phosphorylated AMPK (P-AMPK). Phosphorylated NUAK1 (P-NUAK1). Data: average ± SEM. Statistical test: one-way ANOVA with Dunnett's multiple comparison test. $N_{(DMSO)} = 4$, $N_{(AICAR)} = 4$, $N_{(c-991)} = 4$. ns not significant, *$p \leq 0.05$, **$p \leq 0.01$, ***$p \leq 0.001$. Source data are provided as a Source Data file.

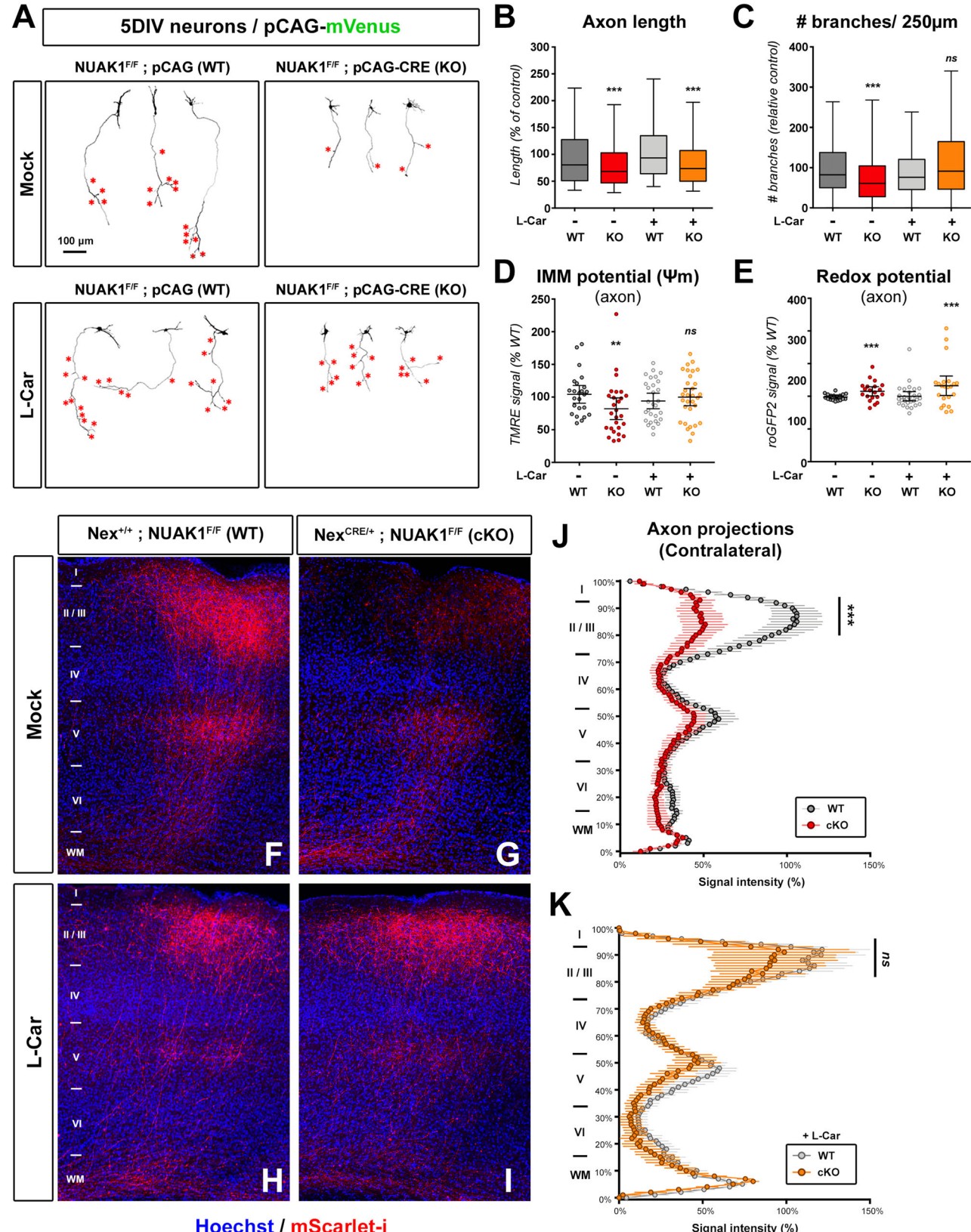

**Figure legend (panels A–K):**

**A** 5DIV neurons / pCAG-mVenus; Mock and L-Car; NUAK1^F/F ; pCAG (WT) and NUAK1^F/F ; pCAG-CRE (KO); scale bar 100 μm.

**B** Axon length; Length (% of control); L-Car − − + +; WT KO WT KO; *** ***

**C** # branches/ 250μm; # branches (relative control); L-Car − − + +; WT KO WT KO; *** ns

**D** IMM potential (Ψm) (axon); TMRE signal (% WT); L-Car − − + +; WT KO WT KO; ** ns

**E** Redox potential (axon); roGFP2 signal (% WT); L-Car − − + +; WT KO WT KO; *** ***

**F, G, H, I** Nex^+/+ ; NUAK1^F/F (WT) and Nex^CRE/+ ; NUAK1^F/F (cKO); Mock and L-Car; Hoechst / mScarlet-i

**J** Axon projections (Contralateral); Signal intensity (%); WT, cKO; ***

**K** + L-Car; Signal intensity (%); WT, cKO; ns

neurons (Fig. S6C–E). In parallel, we measured mitochondrial respiration in neurons treated with vehicle or L-Car (Fig. S7A–C) and observed a normalization of basal and maximal respiration. Developing neurons rely primarily on pyruvate to fuel the TCA cycle and oxidative phosphorylation. Interestingly, L-Car treatment increased both basal and maximal respiration levels even upon inhibition of the mitochondrial

pyruvate transporter using UK5099 (Fig. S7D–F), suggesting that the effect of L-Car rests on the activation of alternative metabolic routes for the mitochondria.

We finally tested if metabolic upregulation can rescue terminal axon branching in vivo by providing L-Car to *Nuak1* cKO mice. As previously demonstrated[17], there is a marked decrease in terminal axon

**Fig. 5 | Upregulation of mitochondrial function rescues axon branching in NUAK1-deficient neurons. A** Representative images of mVenus expressing cortical neurons of *Nuak1*^(f/f) background and electroporated with a control or CRE-coding plasmid. At DIV2 neurons were treated with L-Carnitine (L-Car) (1 mM). Red stars indicate collateral branches. **B**, **C** Quantification of axon length and collateral branches in the indicated conditions. Statistical test: Kruskal-Wallis test with Dunn's post-test (each condition compared to untreated WT condition). $N_{(WT)}$ = 399, $N_{(KO)}$ = 232, $N_{(WT+L-Car)}$ = 387, $N_{(KO+L-Car)}$ = 256 out of 5 independent experiments. Data: box-and-whisker plots. The horizontal line inside the box is the median value. The endpoints of the box represent the 25th−75th percentiles, with whiskers at minimum and maximum values. **D**, **E** Effect of L-Car on the inner mitochondrial membrane potential (IMM, $\Delta\Psi_m$) and redox potential of WT and KO neurons. Each point represents average values for a given neuron. Data: average ± 95% CI. Analysis: unpaired two-tailed *t* test with Welch's correction. **D** $N_{(WT-L-Car)}$ = 29, $N_{(KO+L-Car)}$ = 31 out of 5 independent experiments. **E** $N_{(WT+L-Car)}$ = 28, $N_{(KO+L-Car)}$ = 22 out of 4 independent experiments. **F–I** Contralateral terminal axon branching in *Nuak1* cKO mice following IUCE with a plasmid coding mScarlet-i. The observed decrease in terminal axon branching in KO animals (**G**) was rescued upon L-Car treatment in drinking water (**I**). I–VI indicate cortical layers 1 to 6. WM: white matter. **J**, **K** Quantification of normalized mScarlet-i fluorescence along a radial axis in the contralateral cortex. Data: average ± SEM. Statistical test: two-way ANOVA with Bonferroni's multiple comparison. $N_{(WT)}$ = 8, $N_{(KO)}$ = 6, $N_{(WT+L-Car)}$ = 10, $N_{(KO+L-CAR)}$ = 6 animals. ns not significant, *$p \le 0.05$, **$p \le 0.01$, ***$p \le 0.001$. Source data are provided as a Source Data file.

branching in *Nuak1* cKO mice (Fig. 5F, G, quantified in Fig. 5J). Strikingly, terminal axon branching was indistinguishable between WT and KO mice when mice were treated with L-Car upon detection of gestation (at E13.5) and during lactation (Fig. 5H, I, quantified in Fig. 5K). Taken together, our results demonstrate that the mitochondrial activator L-Car is sufficient to rescue axon branching defects following *Nuak1* deletion, in vitro and in vivo.

### A transcriptomic signature of NUAK1-deficient neurons

We next performed bulk RNA sequencing to uncover the mechanism by which NUAK1 regulates axonal mitochondria. Cortices from constitutive *Nuak1* KO mice were collected at E18.5, corresponding to the latest timepoint when we can collect samples from knockout animals that die at birth (Fig. 6A, 'in vivo'). A first analysis at low stringency (DEGs, Log2FC.threshold >0.25, $p < 0.05$) revealed 453 differentially-expressed genes (Fig. 6B). Over Representation Analyses (ORA) revealed that E18.5 cortices from WT and KO mice have distinct molecular features. Genes over-expressed following *Nuak1* KO (i.e., 267 genes) were associated with protein translation, oxidative processes as well as extracellular matrix production and organization (Fig. 6C and Supplementary Data 1). Interestingly, a large fraction of under-expressed genes (i.e., 186 genes), were related to neuron maturation and metabolism, in particular axonogenesis and synaptic transmission. Altogether, these transcriptional perturbations are compatible with an alteration of cortical neuronal maturation (Fig. 6C). To go further, we performed a second round of RNA sequencing from primary neuronal cultures isolated from E15.5 embryos (Fig. 6A, 'in vitro'). Since neurons were grown in serum-free medium, there was limited astrocytic growth, allowing to focus our analysis more specifically on glutamatergic cortical neurons. Using similar criteria (Log2FC.threshold > 0.25, $p < 0.05$), we identified 336 DEGs (Fig. S8A) and observed a similar ORA signature. Genes under-expressed following *Nuak1* KO (i.e., 214 genes) were again associated with neuronal maturation, including axonogenesis and glutamatergic synapse formation/function (Fig. S8B).

To strengthen these results and further highlight similarities between in vivo and in vitro samples, we performed a Gene Sets Enrichment Analysis (GSEA) that is not dependent on a cutoff to identify DEG, but relies on a ranking of all detected genes (Fig. 6D and Supplementary Data 2). Comparison of gene sets perturbed following *Nuak1* KO highlighted that numerous gene sets were similarly perturbed in both conditions (in vivo and in vitro) (Fig. 6D). Among those, gene sets related to mitochondria appeared particularly interesting, as they suggest perturbation in mitochondrial function (e.g., ATP production, oxidative phosphorylation), in agreement with results presented in this manuscript (Fig. 6E).

### Mitochondrial microprotein BRAWNIN is regulated by NUAK1

To identify key genes similarly perturbed by *Nuak1* KO, we performed a new DEGs analysis, using more stringent criteria (Log2FC.threshold > 0.25, $p < 0.01$). Intersections of results obtained in vivo and in vitro revealed 37 genes, the majority of which (67%) were affected in similar ways (Fig. 7A and Supplementary Data 3). Out of this list, we chose to focus on *Brawnin* (*Uqcc6/C12orf73*), a small open reading frame-encoded peptide that was recently identified as an essential factor for mitochondrial supercomplexes assembly[24,25]. Furthermore, BRAWNIN protein expression is increased in HEK293T cells upon treatment with AICAR[24], which we demonstrated is a potent activator of NUAK1 (see Fig. 4).

*Brawnin* was ranked 3rd in the list of DEGs in both our in vivo (−67%, pAdj 3.16E-10) and in vitro (−66%, pAdj 4.79E-6) analyses. We could confirm a significant decrease of *Brawnin* transcript abundance in total mRNA extracts from mouse embryo cortices (−27,6%) (Fig. 7B) as well as from primary neuronal cultures (−38.7%) (Fig. 7C). While analyzing the expression of *Brawnin* by RT-PCR, we observed a distinct pattern of mRNA splicing with the inclusion of alternative donor sites in the non-coding exon 1, producing three alternative splicing isoforms of *Brawnin* mRNA (Figs. 7D and S9A). The longest form, which we termed variant A, includes the whole sequence of exon 1 (Fig. S9B, C). Two alternative variants included shorter exon 1 and were termed variant B (−129bp) and variant C (−188bp). Strikingly, we observed by RT-PCR a marked decrease of variant A, and to a lesser extent of variant B, and a parallel increase of the shorter variant C, in both cortices and neurons from *Nuak1* KO embryos compared to controls (Fig. 7E, F). Thus, although the coding sequence of *Brawnin* is not affected by the alternative splicing, our data demonstrate that a loss of NUAK1 affects *Brawnin* mRNA abundance and splicing[21].

We tested if BRAWNIN might convey the axonal functions of NUAK1 using an shRNA-based approach in mouse cortical neurons (Fig. S9D). We performed IUCE of two independent shRNA plasmids targeting *Brawnin*. There was no marked deregulation of neurogenesis, neuronal migration and axon formation upon reduction of BRAWNIN level (Fig. 7G–I). Quantification of axon branching suggested a reduction on the ipsilateral layer V at least upon electroporation of shRNA2 of *Brawnin* (Fig. 7M). The reduction of axon branching was more marked on the contralateral side, where either shRNA plasmid targeting *Brawnin* electroporation led to a marked reduction of terminal branching on both layer II/III and layer V, despite similar fluorescence levels in the WM indicating that axon growth and targeting are not affected (Fig. 7J–L, N). Taken together, our results demonstrate that *Brawnin* knockdown phenocopies the branching phenotype seen upon NUAK1 loss.

### BRAWNIN expression is required for cortical axon branching

So far, the neuronal functions of BRAWNIN are unknown. To test the functions of BRAWNIN in developing neurons, we turned to ex vivo cortical electroporations and neuronal cultures allowing to quantify axon morphogenesis and mitochondrial function with a single cell resolution. Inhibition of *Brawnin* expression had little effect on axon growth (Fig. 8A–C). In contrast, we observed a marked reduction of collateral branch formation (Fig. 8D). In parallel, the inhibition of *Brawnin* impaired mitochondrial $\Delta\Psi_m$, to the same extent as *Nuak1* knock out (Fig. 8E). The decrease in BRAWNIN levels resulted in a marked reduction of both basal and maximal respiration (Fig. 8F–H),

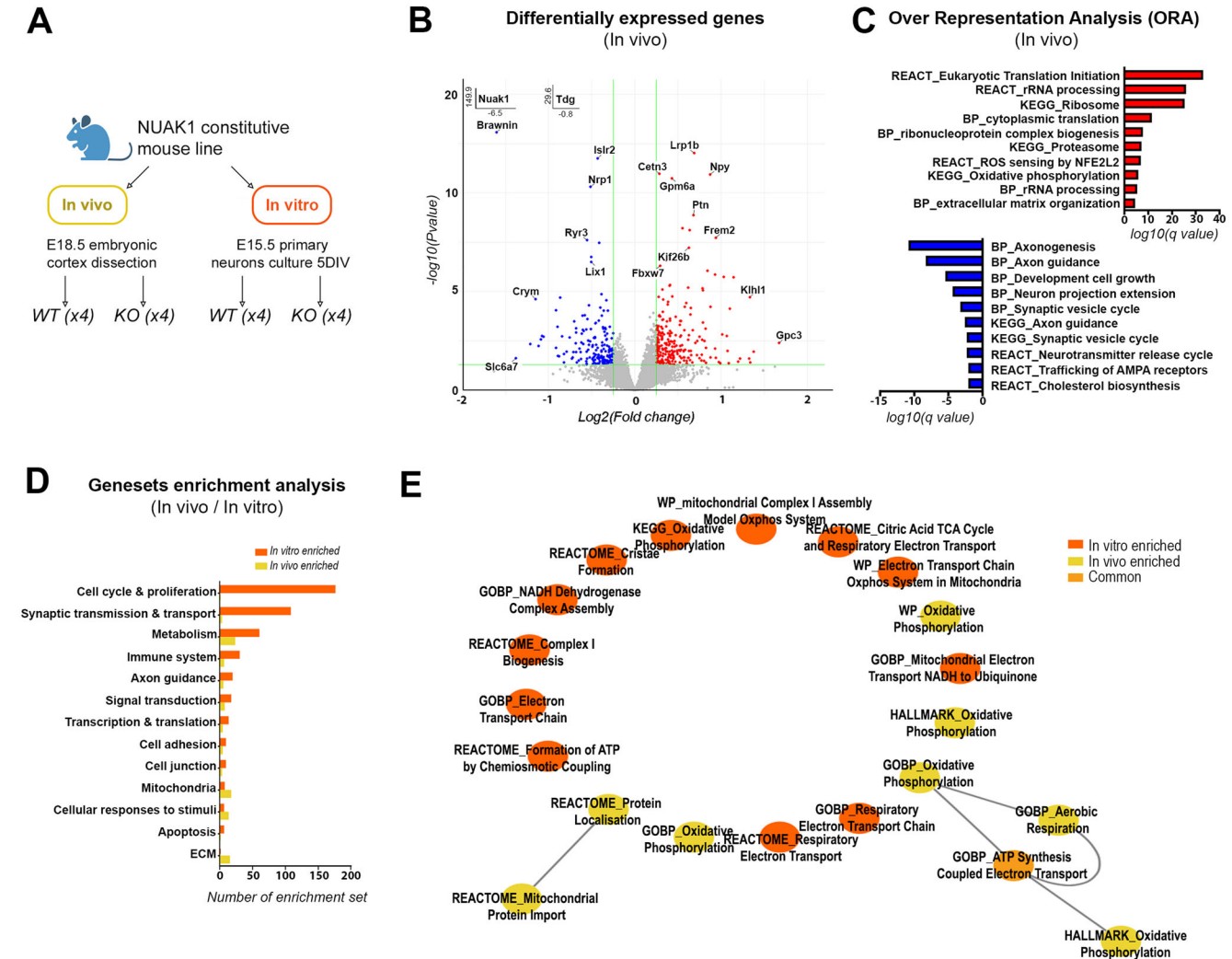

**Fig. 6 | Transcriptomic signature associated to *Nuak1* knockout in cortical neurons. A** Experiment design for in vivo and in vitro transcriptional analysis. Created with BioRender.com. **B** Volcano plot showing the differentially expressed genes (DEG) between *Nuak1* KO and WT cortices (in vivo). Significant DEGs (|log2FC| > 0.25, *p*< 0.05, Wald test) appear in red (upregulated) or in blue (down-regulated). Gene names appear for selected genes in *Nuak1* KO *vs* WT mice. See Supplementary data 1 for complete gene list. **C** Bar plots representing selected over-represented GO BP, KEGG or Reactome categories, associated to transcripts upregulated (red) or downregulated (blue) in E18.5 *Nuak1* KO *vs* WT cortex. **D** Gene set enrichment analysis (GSEA) for *Nuak1* KO vs WT in vivo (yellow) and in vitro (orange). Sets have a false discovery rate (*q* value) < 0.05 and are hand curated into thematic categories to illustrate specificities and similarities of transcriptional changes in both condition. **E** In vitro and in vivo enriched gene sets related to mitochondria metabolism. Source data are provided as Supplementary Data 1, 2.

demonstrating that BRAWNIN is essential to support mitochondrial metabolic activity and collateral branching in developing neurons.

Conversely, we performed rescue experiments by re-expressing BRAWNIN in *Nuak1* knockdown neurons in vitro. As described previously ([11] and this study), the inactivation of *Nuak1* impaired axon development, leading to shorter axons and fewer collaterals at 5DIV (Fig. 8I, J). The overexpression of BRAWNIN had no impact on axon development in wild-type neurons (Fig. 8K). In contrast, BRAWNIN rescued the branching phenotype induced by *Nuak1*-targeting shRNAs (Fig. 8L–N), whereas it failed to rescue axon length, just as was the case with L-Car. The expression of BRAWNIN rescued axonal mitochondria ΔΨ$_m$ in *Nuak1* cKO neurons (Fig. 8O). Overall, our results demonstrate that re-expression of BRAWNIN is sufficient to rescue mitochondria and axon branching phenotypes linked to NUAK1, proving that BRAWNIN is part of a signaling cascade linking NUAK1 to the control of axonal metabolism to support collateral branching in developing axons.

## Discussion

Previous studies demonstrated that a deregulation of mitochondrial positioning, morphology or function disrupts cortical neurons morphogenesis in vitro[39,40], as well as the formation of cortical circuits in vivo[11,41,42]. In the present study, we provide evidence that mitochondria are preferentially recruited to branch-associated synaptic boutons to support branch stabilization, presumably through metabolic remodeling at the nascent synapse. Furthermore, we demonstrate for the first time that the ASD-linked kinase NUAK1 regulates mitochondrial metabolic activity within cortical neurons axons, through a novel effector, BRAWNIN, whose neuronal functions were previously unknown. Since BRAWNIN is involved in mitochondrial respiratory chain assembly[24,25], our results support a role of BRAWNIN in mediating the effects of NUAK1 on the regulation of mitochondrial activity[20,21].

Our experiments demonstrate that in mouse cortical neurons, axonal branches are formed first, and that a subsequent recruitment of mitochondria to presynaptic boutons stabilizes nascent branches. This model is compatible with previously described modalities of axon branching in Retinal Ganglion Cells (RGCs), where axonal branches originate from pre-existing presynaptic sites and are later stabilized by the formation of a novel presynaptic site[43–45]. Yet, this differs from observations in Peripheral Nervous System (PNS)

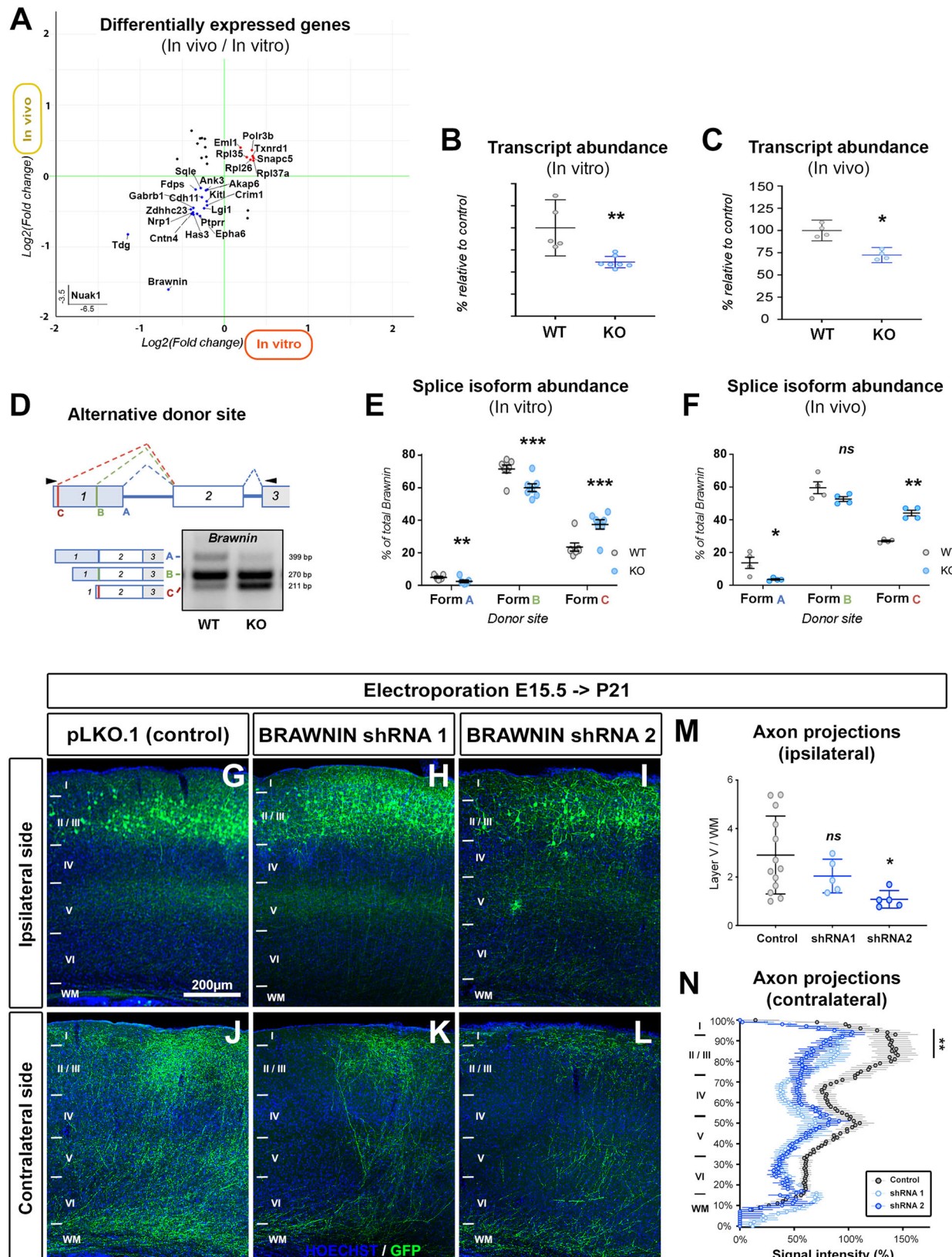

neurons such as sensory neurons, where mitochondria recruitment to hot-spots of local translation is a prerequisite to the formation of axonal branches[6,46]. Although we do not rule out a possible role of NUAK1 and mitochondrial recruitment in supporting local protein synthesis in cortical neurons, it is likely that this discrepancy stems from distinct modalities of collateral branch formation

within neurons of the CNS and PNS. Furthermore, the relevance of NUAK1 for PNS neuron development has not been investigated. Interestingly, studies suggest other AMPK-RKs can functionally operate in regulating axon branching in PNS neurons[47], potentially explaining mechanistic differences in the modalities of branch formation.

**Fig. 7 | The microprotein BRAWNIN regulates cortical axon branching in vivo.**
**A** Graph comparing Differentially Expressed Genes (|log2FC| > 0.25, $p < 0.01$, Wald test) upregulated (red) and downregulated (blue) in vivo or in vitro following *Nuak1* KO. **B-C** RT-qPCR validation of *Brawnin* mRNA downregulation within the cortex of E18.8 *Nuak1* KO mice (**B**), as well as in primary neuronal cultures (**C**). $N_{(WT)} = 5$, $N_{(KO)} = 7$ (**B**). $N_{WT} = 4$, $N_{KO} = 4$ (**C**). Statistical test: two-tailed Mann-Whitney. Data: average ± SEM. **D** Schematic representation of the splicing donor sites of *Brawnin* transcript and corresponding splicing isoforms, and splicing profile in WT and *Nuak1* KO primary neurons (**E**) and cortices (**F**). Each form is represented as a percentage of total amount of *Brawnin* transcripts. $N_{(WT)} = 7$, $N_{(KO)} = 7$ (**E**). $N_{(WT)} = 4$, $N_{(KO)} = 4$ (**F**). Statistical test: two-way ANOVA with Bonferroni's multiple

comparison (**E**, **F**). Data: average ± SEM. bp: base pairs. **G–L** Coronal sections of mouse brains following *in utero* electroporation of shRNAs against *Brawnin*. mVenus expression was used to visualize electroporated neurons. I–VI indicate cortical layers 1 to 6. WM: white matter. **M** Quantification of normalized mVenus fluorescence on Layer V on the ipsilateral. Data: average ± SEM. Statistical test: one way ANOVA with Bonferroni's multiple comparison. $N_{Control} = 13$, $N_{shRNA1} = 6$, $N_{shRNA2} = 5$ animals. **N** Quantification of normalized mVenus fluorescence along a radial axis in the contralateral cortex. Data: average ± SEM. Statistical test: two-way ANOVA with Bonferroni's multiple comparison. $N_{(Control)} = 13$, $N_{(shRNA1)} = 4$, $N_{(shRNA2)} = 6$ animals. ns not significant, $*p \leq 0.05$, $**p \leq 0.01$, $***p \leq 0.001$. Source data are provided as a Source Data file and Supplementary Data 3.

Our results suggest that the recruitment of mitochondria to immature presynaptic boutons plays a specific role for synapse maturation, which supports branch stabilization. Although we did not observe changes in the levels of proteins involved in mitochondrial dynamics, we cannot exclude the possibility that NUAK1 plays a role in maintaining a pool of healthy mitochondria at the synapse by controlling mitochondrial turnover. Synapses account for the majority of ATP consumption in neurons[48–50], and presynaptic mitochondria support synaptic function through a crosstalk between ATP-producing oxidative phosphorylation[50,51] and calcium buffering[52–55]. Although the metabolic specificities of immature presynaptic boutons remain largely unknown, it is unlikely that ATP by itself is sufficient to support synapse maturation and branch stabilization. Indeed, the metabolic functions of mitochondria are not restricted to ATP production but contribute to most metabolic pathways in the cell including the metabolism of lipids, amino acids, nucleic acids or neurotransmitters[56]. Furthermore, axon outgrowth and branching are dependent on local calcium dynamics[57] and calcium buffering by mitochondria is important for cortical PN axons branching[41]. Since mitochondrial ATP production and calcium dynamics are intertwined[51,58], it is possible that impaired calcium dynamics at nascent presynaptic boutons contribute to the branching phenotype reported in *Nuak1* KO neurons, as demonstrated for the NUAK1 upstream kinase LKB1[54,59]. Further studies are needed to carefully dissect the metabolome of mitochondria captured at axonal branchpoints in developing axons.

One striking observation from our study is the fact that axon elongation and collateral branching are two distinct biological processes supported by distinct metabolic modalities. Firstly, axon elongation is directly dependent upon D-glucose concentration suggesting it is directly correlated to non-oxidative metabolism pathways, whereas collateral branching is supported in an all-or-nothing fashion by directly fueling mitochondrial metabolism. Secondly, BRAWNIN knockdown impairs axon branching without affecting axon elongation. Finally, upregulation of mitochondrial metabolism through L-Car and vitamin K2, or through re-expression of BRAWNIN, can rescue axon branching in a *Nuak1* null background, without significantly affecting axon length. This data, together with the selective effect of an autism-linked truncated *NUAK1* mutant in axon branching but not axon elongation[17], converge in supporting an essential role for mitochondrial metabolism in collateral branching. Future work will determine what other metabolic pathways support axon elongation, and whether these are also dependent upon NUAK1 function.

Other studies in developing neurons suggest that AMPK activation with AICAR promotes PGC-1α-dependent mitochondria biogenesis[60], as well as mitochondria transport and axon branching[29]. In mature neurons, it was reported that synaptic ATP depletion induces local AMPK activation and mitochondria capture at presynaptic boutons to restore energy homeostasis[9]. Furthermore, pharmacological activation of AMPK was reported to increase axon branching in dentate granule cells[29]. These observations, based on the use of non-selective AMPK activator AICAR or the promiscuous kinase inhibitor Compound-C (also known as Dorsomorphin), contradicting the fact that the genetic inactivation of *Ampk*, by knockout of the two catalytic isoforms (α1/α2)[10] or

the regulatory subunit ß1[61] does not lead to apparent cortical development defects. Here, we demonstrate that AMPK is not required for axonal elongation and branching in cortical PNs in vitro and in vivo, suggesting that its functions might be restricted to more mature neurons. Rather, we observe that AICAR is equally potent at activating NUAK1. Although we did not demonstrate that AICAR (ZMP) acts directly on NUAK1, it is unlikely that its effects are mediated through an activation of the upstream kinase LKB1, since previous studies established that LKB1 is not altered in response to AICAR in mouse embryonic fibroblasts[35] or rat skeletal muscles[62]. Notably, Compound-C and SBI-0206965, which are commonly used as AMPK inhibitors, are also potent inhibitors of NUAK1[63,64]. Thus, our work raises caution into interpreting the effects of non-selective AMPK activators or inhibitors in immature neurons, and suggests that some of the effects attributed to AMPK may instead be caused by NUAK1. Future studies are warranted to investigate the mechanism by which AICAR activates NUAK1 and screen for small-molecule NUAK1 selective activator.

We present evidence that NUAK1 regulates mitochondrial metabolism at least in part through the regulation the mitochondrial microprotein BRAWNIN. The discovery of numerous mitochondrial microproteins has contributed to deciphering the process of assembly and function of respiratory chain supercomplexes. Current literature has now demonstrated the importance of such proteins not only in mitochondrial function, but also in larger physiological processes such as unfolded protein response, activation of innate immunity and cardiac activity[65]. Furthermore, a DNA variant of the coding sequence of one such protein, SHMOOSE, has been associated with Alzheimer's disease[66] implicating this family of proteins in neurodegeneration. Despite the important advances in the field, data concerning the involvement of mitochondrion-located peptides in neurodevelopmental mechanisms remains scarce. Our study shows for the first time the crucial role of a mitochondrial microprotein (BRAWNIN) in the development of cerebral cortex and neural circuits. Furthermore, re-expression of BRAWNIN is sufficient to restore mitochondrial function and axon branching in NUAK1-deficient neurons, opening potential avenues for early therapeutic intervention in neurodevelopmental disorders by modulating the levels of mitochondrial microproteins[67].

Finally, our work provides molecular insight into the mechanisms of metabolic regulation in developing neurons, opening perspective for future targeted therapeutics in neurodevelopmental disorders. Mounting evidence supports that altered mitochondrial metabolism is linked to an increased risk of various neurodevelopmental disorders including Autism Spectrum Disorders[68,69]. Mitochondrial proteins are over-represented in protein interaction network of ASD risk genes[70]. Mitochondrial defects can cause an increased ROS production, which has been linked to defects in cortical circuit formation[42]. The use of ROS scavenger can alleviate some of the anatomical and behavioral phenotypes in a mouse model of schizophrenia[71]. However, we did not observe any benefit of N-AC treatment in NUAK1-deficient neurons. On the contrary, the fact that L-Car rescued axonal branching in vitro and in vivo demonstrates the importance of mitochondrial metabolic function for cortical circuits development. Especially, L-Car has demonstrated some beneficial effects on axonal development and branching in sensory

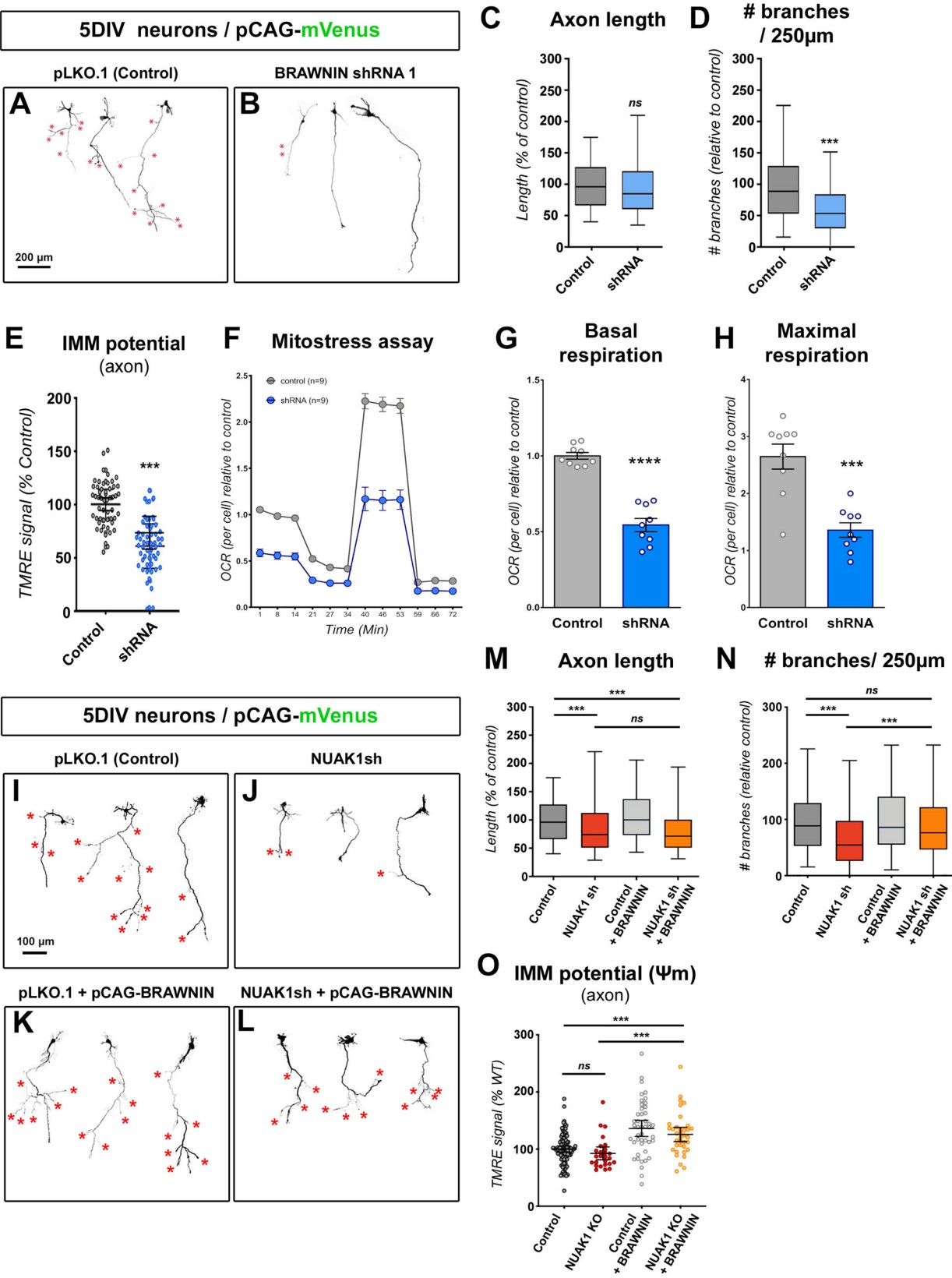

neurons[36]. Furthermore, L-Car has been shown to be neuroprotective in a wide variety of metabolic stresses[72,73] and has clinical interest in the treatment of depression and anxiety[74,75] by an upregulation of metabolic activity[75,76]. Our work provides further evidence that L-Carnitine might be of therapeutic interest in correcting mitochondrial dysfunction and metabolic imbalance during neurodevelopment.

## Methods

### Primary neuronal culture and Ex Vivo Cortical Electroporation

The electroporation of dorsal telencephalic progenitors was performed by injecting plasmid DNA (1–2 µg/µL of endotoxin-free plasmid DNA) plus 0.5% Fast Green (Sigma; 1:20 ratio) using a Picospritzer III microinjector (Harvard Apparatus) into the lateral ventricles of

**Fig. 8 | BRAWNIN is necessary and sufficient downstream of NUAK1 for cortical axons branching. A**, **B** Representative images of mVenus expressing cortical neurons electroporated with an shRNA against *Brawnin*. Red stars indicate collateral branches. **C**, **D** Quantification of axon length and collateral branches in the indicated conditions. Statistical test: two-tailed Mann-Whitney. $N_{(control)}$= 295, $N_{(shRNA)}$= 277 neurons out of 3 independent experiments. Data: box-and-whisker plots. The horizontal line inside the box is the median value. The endpoints of the box represent the 25th−75th percentiles, with whiskers at minimum and maximum values. **E** Effect of *Brawnin* shRNA on the mitochondrial membrane potential ($\Delta\Psi_m$). Each point represents average values for a given neuron. $N_{(control)}$= 51, $N_{(shRNA)}$= 53 out of 3 distinct cell cultures. Data: average ± SEM. Statistical test: two-tailed Mann-Whitney. **F**–**H** Measurement of oxygen consumption rate in 5DIV cortical neurons transduced with shRNA against *Brawnin*. OCR values are normalized to cell number. Basal and maximal respiration values are represented in (**G**) and (**H**) respectively. Data: average ± SEM. $N_{(control)}$ = 9, $N_{(shRNA)}$ = 9 different embryos out of 3 independent litters. Statistical test: two-tailed Mann-Whitney. **I**–**L** Representative images of cortical neurons electroporated with an shRNA against *Nuak1* and overexpressing BRAWNIN. Red stars indicate collateral branches. **M**, **N** Axon length and collateral branches of 5DIV neurons in the indicated conditions. Statistical test: Kruskal-Wallis test with Dunn's post-test. $N_{(Control)}$= 295, $N_{(NUAK1sh)}$= 302, $N_{(Control+BRAWNIN)}$ = 277, $N_{(NUAK1sh+BRAWNIN)}$ = 316 neurons out of 3 independent experiments. Data: box-and-whisker plots. The horizontal line inside the box is the median value. The endpoints of the box represent the 25th−75th percentiles, with whiskers at minimum and maximum values. **O** Measurements of the inner mitochondrial membrane potential (IMM) in the indicated conditions. Data: average ± 95% CI. Statistical test: Kruskal-Wallis test with Dunn's post-test (each condition compared to the control condition). Data: average ± 95% CI. $N_{(control)}$= 97, $N_{(NUAK1\ KO)}$ = 26, $N_{(control+BRAWNIN)}$ = 45, $N_{(NUAK1\ KO+BRAWNIN)}$ = 36 neurons out of 4 independent experiments. ns not significant, *$p \leq 0.05$, **$p \leq 0.01$, ***$p \leq 0.001$. Source data are provided as a Source Data file.

isolated E15.5 embryonic mouse heads. Electroporations were performed with gold-coated electrodes (GenePads 5 mm, BTX) using an ECM 830 electroporator (BTX) and the following parameters: five pulses of 100 milliseconds (ms), 150 ms interval, at 20 V. Immediately after electroporation, cortices were dissected in Hank's buffered salt solution (HBSS) supplemented with HEPES (pH 7.4; 2.5 mM), $CaCl_2$ (1 mM, Sigma), $MgSO_4$ (1 mM, Sigma), $NaHCO_3$ (4 mM, Sigma), and D-glucose (30 mM, Sigma), hereafter referred to as complete HBSS (cHBSS). Isolated cortices were dissociated in cHBSS containing papain (Worthington, 20 μ/mg at least) for 20 min at 37 °C. Cortices were washed once in cHBSS containing DNase I (2.5 mg/mL, Sigma), then 3 times in cHBSS before being dissociated. Cells were then plated at $125.10^3$ cells per 35 mm glass bottom dish (MatTek) coated with poly-D-lysine (0.1 mg/mL) and laminin (0.01 mg/mL) and cultured for 5−7 days in Neurobasal medium supplemented with B27 (1x), N2 (1x), Glutamax (2 mM), and penicillin (10 μ/mL)-streptomycin (0.1 mg/mL).

## In utero cortical electroporation
In Utero Cortical Electroporation were performed at E15.5 as described in[30]. A mix containing 1 μg/μl endotoxin-free plasmid DNA plus 0.5% Fast Green (Sigma; 1:20 ratio) was injected into one lateral hemisphere. Electroporation was performed using an ECM 830 electroporator (BTX) using four pulses of 45 V with 500 ms interval to target cortical progenitors. Animals were sacrificed at Postnatal day 21 by terminal perfusion of 4% paraformaldehyde (PFA, Electron Microscopy Sciences) followed by overnight post- fixation in 4% PFA.

## Immunohistochemistry
75 μm thick sections were performed using a Leica VT1000S vibratome. Slices were permeabilized for 30 min in Permeabilization Buffer (PB, PBS 1x, BSA 0.5%, Triton X-100 0.1%), then incubated overnight with primary antibodies (polyclonal chicken anti-GFP, Rockland) diluted at 1:2000 in PB. The following day, we performed 3 washes in PBS 1X, then incubated slices in secondary antibody-containing PB (polyclonal goat anti-chicken antibody, Alexa 488, 1:2000, Life technology). Nuclear DNA was stained using Hoechst 33258 (1:5000, Pierce).

## Cell lines, culture conditions, transfections, and drug treatments
Unless indicated otherwise, drugs were from Sigma-Aldrich and diluted in water. For in vitro rescue experiments, neurons were treated at 2 DIV with either Acetyl-L-Carnitine (1 mM), Vitamin K2 (Menaquinone K4) (1 μM in ethanol), Vitamin K2 (Menaquinone K3) (1 μM in ethanol), N-Acetyl-L-Cystein (1 mM).

For measurements of mitochondrial Inner Membrane potential, neurons were pre-incubated in cHBSS for 30 min at 37 °C, then incubated in cHBSS containing 20 nM of TMRE for 30 min at 37 °C. Medium was replaced by cHBSS containing 5 nM of TMRE immediately before live-imaging. FCCP (2 μM) was added after 5 min of imaging.

Human HeLa (ATCC CRL-1772) and HEK293T (ATCC CRL-3216) cells were grown at 37 °C under 5% CO2 in DMEM supplemented with 10% FBS. HeLa cell transfections were performed with pEGFP-Flag-mNUAK1 and pCIG2-mLKB1 using JetPrime-Polyplus reagent according to the manufacturer's instructions. Cells were used for experiments 48 h after transfection. AICAR and c-991 were used at a final concentration of 1 mM and 10 mM respectively for 1 h. HEK293T cells were transfected with a control pLKO.1 vector or pLKO shRNA1 and shRNA2 against mouse *Brawnin* and pCAG-mBRAWNIN-V5tag. Cells were lysed 72 h after transfection.

## Image acquisition and analyses
Confocal images were acquired in 1024 × 1024 mode with a Nikon Ti-E microscope equipped with the C2 laser scanning confocal microscope. Time-lapse images were acquired in 1024 × 1024 mode using a CMOS ORCA-Flash 4.0 Camera (Hamamatsu). Microscope control and image analysis was performed using the Nikon software NIS-Elements (Nikon). We used the following objective lenses (Nikon): 10x PlanApo; NA 0.45, 20x PlanApo VC; NA 0.75, 60x Apo; NA 1.4. When live-cell imaging was carried out, the temperature was maintained at 37 °C and CO2 at 5% v/v in a humidified atmosphere. Live-cell imaging was performed in cHBSS. For live imaging of mitochondria, we detected 45 events of spontaneous branch formation over 24 neurons and quantified mitochondria localization at branchpoints and within branches, as well as axonal branch length and lifetime.

For axonal morphogenesis experiments, we performed large field acquisition of neuronal cultures (typically a 4 × 4 tiling scan using a 20x objective). All neurons on the reconstituted image were quantified and axon length was measured with the Nikon NIS Elements software. Axons shorter than 80 μm were not counted. To make figures, representative neurons were isolated from the rest of the image using ImageJ. Contrast was enhanced and background (autofluorescence of no transfected neurons in culture) removed for better illustration of axons morphology. Kymographs were created with NIS-Elements.

To assess putative defects in terminal axon branching in vivo, we measured GFP fluorescence intensity and normalized to GFP fluorescence intensity in the axons of the WM, which is correlated to electroporation efficiency. On the ipsilateral side, 5 ROIs were drawn in the axonal plexus on layer 5 using NIS-Elements. The same ROIs were then duplicated and placed in the corpus callosum (for normalization), and in the thalamus (background fluorescence). Signal was defined as (Average (layer 5) − Average (background)) / (Average (corpus callosum) − Average (background)).

On the contralateral side, we measured the signal intensity along a line from the ventricular zone to the pial surface using the profile plot function of ImageJ (width: 64 pixels on each side of the line). We used an excel spreadsheet to perform background subtraction and binning of data along the ventricular zone to pial axis (from 0 to 100%). Data was normalized by the quantity of signal in the white matter, which

indicates the quantity of axons reaching the contralateral cortex. Briefly, the excel spreadsheet determined the maximum signal intensity value in the 0–20% bins (corresponding to the white matter), and this value was considered as 100%. Using this method, we could ensure that our macro detected changes in the branching pattern, regardless of the quantity of axons reaching the contralateral hemisphere.

## Statistical analyses
Statistical analyses were performed using Prism (GraphPad). Statistical tests and number of replicates are indicated in figure legends. Quantifications were performed blind to genotype.

## Reporting summary
Further information on research design is available in the Nature Portfolio Reporting Summary linked to this article.

## Data availability
Data used to generate all quantification results presented in the figures of this manuscript are available upon reasonable request to the corresponding author. Source data are provided with this paper as a Source Data file. Transcriptomic data presented in Figs. 6, 7 and Supplementary Fig. 8 has been deposited on GEO. · RNAseq data ("in vitro") can be accessed on GEO with accession code GSE227203 · RNAseq data ("in vivo") can be accessed on GEO with accession code GSE226698. Source data are provided with this paper.

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

## Acknowledgements

The authors thank members of the Courchet lab and Institut Neuro-MyoGène for useful comments and discussion. We are sincerely grateful to Franck Polleux for his support in building this project.

We wish to thank Franck Polleux, Tommy Lewis, Yusuke Hirabayashi, Seok-Kyu Kwon, Hélène Puccio and Thomas Boulin for technical advice, suggestions, and critical reading of the manuscript, and Agnès Duplany, Anne Devin and Arnaud Mourier for suggestions and technical help in setting up biochemical assays. We thank the personnel from the SCAR and ALECS-SPF mouse facility for animal care. We acknowledge support from the PSMN (Pôle Scientifique de Modélisation Numérique) of the ENS of Lyon for computing resources. Sequencing was performed by Gregory Amann at the GenomEast platform, a member of the 'France Génomique' consortium (ANR-10-INBS-0009). This work was supported by the Fondation pour la Recherche Medicale (AJE20141031276, J.C.) and ERC Starting Grant (678302-NEUROMET, J.C.). This work was performed within the framework of the LABEX CORTEX (ANR-11-LABX-0042 / ANR-11-IDEX-0007, J.C.). M.L. was the recipient of a grant from AFM-telethon through the strategic MyoNeurALP alliance. K.S. is supported by the grant from Novo Nordisk Foundation (NNF18CC0034900).

## Author contributions

M.L. performed in vitro experiments on mitochondria capture at presynapses, impact of energy source and glucose concentration on axon development, effect of AICAR and c-991 on mitochondria-presynapse association, and in vitro effects of L-Car. S.Y. performed seahorse analyses, and in vitro experiments on BRAWNIN, analyzed and interpreted data, and contributed to manuscript writing. J.C. and G.M-D. performed and interpreted IUCE experiments. S.E. and O.R. performed bioinformatic analyses. M.K. set up and optimized the use of lentiviruses. R.D.R. performed mtKR experiments. A.G. assisted M.L. for neuronal cultures. C.B. analyzed long-time lapse films of mitochondria distribution at axonal branches. A.A. performed enzymatic measurements of CS activity. E.G. performed western blot experiments and analyses of BRAWNIN expression and splicing. D.P., H.P. and C.F.B. analyzed transcriptomic datasets. A.K. and R.M. contributed to the analysis of neuronal AMPK. M.F. and B.V. provided conditional AMPK KO mice. K.S. contributed to analysis of NUAK1 regulation by pharmacological agents. J.C. initiated the work, performed metabolic measurements in NUAK1 KO neurons using biosensors, supervised the work and analyzed data. JC drafted the manuscript and all other authors edited and approved the manuscript.

## Competing interests

A patent application EP 23 305 322 entitled "BRAWNIN AGONISTS FOR USE IN THE TREATMENT OF AXONAL METABOLIC DISORDERS" was filed on March 9, 2023 (S.Y. and J.C.). The remaining authors have no competing interests.
