## [Peer Review File · Nature Communications]

The AMPK-related kinase NUAK1 controls cortical axons branching by locally modulating mitochondrial metabolic functionsREVIEWER COMMENTS

Reviewer #1 (Remarks to the Author):

Lanfranchi and Yandiev, et al. have characterized that NUA1 controls cortical axon branching by facilitating mitochondrial metabolism at the branching site through the supercomplex assembly peptide BRAWNIN. Their manuscript is well written, meticulously conducted, and uncovers new, mechanistic ground in the role that mitochondria play in axon branching. I have two points for the authors to consider so to strengthen their manuscript.

1. The authors conclude on P7, Lines 26-29 that the lack of change in mtDNA content in NUA1 KO cultures relative to WT conditions rules out that the decreased respiratory rate is a secondary consequence of impaired mitochondria biogenesis. Measures of mitochondrial content (e.g. mitochondrial protein expression and mtDNA) reflect the sum total of biogenesis and breakdown (i.e. mitochondrial turnover). If mitochondrial turnover uniformly increases or decreases between different models, that is to say equivalent changes in biogenesis and breakdown, measures of mitochondrial content, such as mtDNA, would still be equivalent, even though the biological implications between these two conditions would be quite different. Given the significant decline in respiration and citrate synthase activity in NUA1-deficient neurons (Fig. 3B-D and F), would not a more probable interpretation of the data be that NUA1 supports increased mitochondrial turnover at branch points, thus maintaining their function? This may explain why forcing increased mitochondrial metabolism through L-Car bypasses the requirement of NUA1 and why scavenging ROS (a consequence of impaired turnover) is not sufficient to rescue the branching phenotype. Ideally, direct measures of mitochondrial turnover via isotope labeling are needed or a more thorough characterization of markers of mitochondrial turnover (biogenesis and breakdown).

2. Given the metabolic requirement of mitochondria for branching that the authors characterized so well, it was curious to me why the authors did not go further with their studies of BRAWNIN over- and under-expression. Does inhibiting BRAWNIN impair mitochondrial respiration similar to NUA1 KO (Fig. 3)? Does reintroducing BRAWNIN in NUA1 KO cells maintain mitochondrial respiration?

Minor

- P8, Lines 13, 15: lower case “l” is used in the text in contrast to the upper case “L” in the figure.
- Based on the text on P12, L14-6, are there no statistical significance to report in Fig. 6 I and J?

Reviewer #2 (Remarks to the Author):

This study offers a powerful argument that the Nauk1 kinase is fundamental in safeguarding appropriate mitochondrial metabolism in branching axonal collaterals of cortical neurons. It is strongly suggested

that this function is carried out, at least in part, through the expression and splicing of the BRAWNIN protein in maintaining appropriate mitochondrial functions during the morphogenesis of cortical neurons.

The nuanced dance of activation of mitochondrial metabolism, affecting cell morphogenesis and cell biology, is a known phenomenon in this and related systems. However, what sets this research apart is the outstanding caliber of work, the significant relevance of neurodevelopmental disorders, and the exciting potential for these findings to be applied to other brain areas. In particular, the carefully designed experiments, the interweaving of in vivo data, and the well-executed epistasis experiments were very appealing. In the realm of neural development, this study is a noteworthy and praiseworthy addition.

In the face of numerous experiments that naturally invites the possibility to level a degree of critique, this reviewer chooses to refrain from nitpicking. The rationale behind this decision is the steadfast belief that minor critiques would hardly alter the robust core message presented in the paper. Rather than frittering away resources on minor adjustments, it would be far more beneficial for the research team to channel these toward their forthcoming studies.

Reviewer #3 (Remarks to the Author):

In this manuscript, the authors study the role of mitochondrial trafficking in axon branching, focusing on the role of NUA1, a kinase they previously found to be involved in these processes. They first observe that mitochondrial stalling in axons is associated with stabilisation of new branches as the mitochondria enter the branches, rather than with the initial branch formation. Next, they study differential requirements for AMPK and NUA1, and find no requirement for AMPK in their measures of mitochondrial stalling and axon branching, in vivo and in primary neuron cultures. In parallel, they confirm previous observations that AICAR, a widely used AMPK activator, is not specific for AMPK, but also activates the AMPK-like NUA1. Finally, based on RNA-seq from WT and NUA1-mutant cortices and neuronal cultures, they identify Brawnin, a mitochondrial localised peptide, as a key mediator of NUA1-induced mitochondrial activity in axons, and subsequent branching phenotypes.

This is a solid manuscript, with really interesting and novel findings, providing significant new insight into how mitochondrial metabolism and trafficking interact with axon branching and how this is regulated. The manuscript is well written, with clear and convincing figures, and the experiments seem overall well conducted, with different types of experiments agreeing and supporting the conclusions made throughout.

I am supportive of publication, but have several concerns, mostly minor, listed below, that should be addressed first.

1. Some of the conclusions are overstated, or imply causality when the experiments merely show a correlation. Examples:

- Page 5 line 24: 'branches that failed to recruit rapidly retracted', while the experiments show a correlation in time.

- Page 8 line 19-21: 'a function that can be uncoupled from its role in trafficking'. It's not clear to me how this can be stated, without rescuing mitochondrial trafficking in NUA1 mutant neurons. This experiment is not required for publication, but the interpretation is not correct.

- Page 12 line 17-20: the experiments do not address direct impact of NUA1 on Brawnin splicing, and differential splicing could have many alternative explanations.

2. Fig. 1O,P: comparison and statistical analysis should be done on +/- mitoKillerRed as well (i.e. comparing O vs P, ideally in the same graph).

3. Fig. 3A-D: How come the mtDNA content is not different, if the axonal length and branch number are so much lower in the NUA1 KO neurons? Where have the mitochondria gone then? Is the soma larger, or are there mitochondria per cell volume?

4. Fig. 3I: What is the source of the increased ROS production if the OxPhos activity is significantly decreased? The fig references on page 8 are not always correct (eg I instead of M).

5. Fig. 3H-K: I could not find how the Perceval signal was quantified and normalised. Values from these genetically encoded sensors can be highly variable and sometimes difficult to interpret. The authors should show the raw (non-normalised) values in a supplementary figure, and explain better what is meant by the F0/Fmin values in Fig. 3.

6. Fig. 4F-H: the authors should address (or discuss) the possibility that AICAR may act on LKB1, upstream to NUA1, rather than on NUA1 directly.

7. Page 9-10 and Fig. 5 are interpreted too optimistic.

- The statement that L-Car is a 'mitochondria modulator', and upregulates mitochondrial function by promoting LCFA metabolism is an over-simplification. Axonal OxPhos activity could well be driven by (lactate-derived?) pyruvate and TCA-cycle activity, rather than FA oxidation.

- It is not clear how increasing FA metabolism by LCAR (if this is indeed what happens) can increase OxPhos activity if Fig. 3 already demonstrates decreased maximal respiration?

- I am not convinced that the quantification of contralateral axonal density after IUE is an appropriate measure of axon branching. This is also the case for Fig. 7F-M.

The authors should either restrict their interpretations to description of the experiments, or should perform additional experiments: (1) measure mitochondrial OxPhos activity and PDH phosphorylation upon LCAR and MK 4 treatment; (2) overexpress or inactivate LDHA vs B to activate lactate/pyruvate-driven TCA-cycle activity; (3) quantify axonal density in the corpus callosum as a measure of axon length, and explain how variability between different IUE experiments is accounted for.

8. Fig. 7R,S,T: NUA1KO/sh + Brawnin should be compared to NUA1KO/sh alone as well.

Rebuttal letter for article NCOMMS-23-13021

We would like to thank the reviewers for their overall level of enthusiasm towards our results and for their constructive suggestions. We have addressed most of their points with new data, or modified the text according to their suggestions. We are convinced that the revised manuscript is significantly improved as a result.

Reviewers' comments:

Reviewer #1 (Remarks to the Author):

Lanfranchi and Yandiev, et al. have characterized that NUAK1 controls cortical axon branching by facilitating mitochondrial metabolism at the branching site through the supercomplex assembly peptide BRAWNIN. Their manuscript is well written, meticulously conducted, and uncovers new, mechanistic ground in the role that mitochondria play in axon branching. I have two points for the authors to consider so to strengthen their manuscript.

1. The authors conclude on P7, Lines 26-29 that the lack of change in mtDNA content in NUAK1 KO cultures relative to WT conditions rules out that the decreased respiratory rate is a secondary consequence of impaired mitochondria biogenesis. Measures of mitochondrial content (e.g. mitochondrial protein expression and mtDNA) reflect the sum total of biogenesis and breakdown (i.e. mitochondrial turnover). If mitochondrial turnover uniformly increases or decreases between different models, that is to say equivalent changes in biogenesis and breakdown, measures of mitochondrial content, such as mtDNA, would still be equivalent, even though the biological implications between these two conditions would be quite different. Given the significant decline in respiration and citrate synthase activity in NUAK1-deficient neurons (Fig. 3B-D and F), would not a more probable interpretation of the data be that NUAK1 supports increased mitochondrial turnover at branch points, thus maintaining their function? This may explain why forcing increased mitochondrial metabolism through L-Car bypasses the requirement of NUAK1 and why scavenging ROS (a consequence of impaired turnover) is not sufficient to rescue the branching phenotype. Ideally, direct measures of mitochondrial turnover via isotope labeling are needed or a more thorough characterization of markers of mitochondrial turnover (biogenesis and breakdown).

Response:

We thank this reviewer for this excellent suggestion. Indeed, NUAK1 could affect axonal mitochondrial dynamics, from biogenesis to elimination. As a matter of fact, changes in mitochondria biogenesis or fusion/fission impact axon branching¹⁻³. Furthermore, NUAK1 has been linked to mitophagy in developing muscles in the fly model⁴.

To address this experimentally, we first tried to implement the use of MitoTimer⁵ to follow mitochondrial protein half-life in the axon. MitoTimer is a mitochondria-targeted fluorescent protein that shifts from green to red as the fluorophore matures, thus providing a way to track nascent protein expression. A tetracycline-inducible promoter provides temporal control over protein expression, allowing to correlate green/red fluorescence to protein synthesis and elimination. Unfortunately, preliminary tests in control neurons were inconclusive. After several attempts, we only achieved weak labelling of mitochondria and could not get consistent and reliable results (see **Figure 1 to reviewers**). In some neurons we observed fluorescence prior to doxycycline treatment, while in other neurons there was no fluorescent protein expression even after doxycycline treatment. Without the possibility to control the timing of expression of MitoTimer, we could not correlate fluorescence with protein turnover. Although we will continue to implement this method for future studies, we feel this development will take several more months.

To circumvent this issue, we performed Western-blot analyses in mouse cortices from NUAK1 cKO mice. There was no difference in the expression of proteins involved in mitochondria biogenesis (PGC1 α) or fusion/fission (DRP1, OPA1, MFN1 and MFN2) between WT and KO samples. There were no changes in levels of LC3 or in the phosphorylation of ubiquitin (Ser65), which is typically used as a marker for PINK1 activity (known to be involved in the process of mitophagy). Although these experiments do not rule out a local phenomenon (e.g. branchpoints as suggested), our interpretation is that NUAK1 cKO does not induce major changes in the

regulation of mitochondrial dynamics in the mouse cortex. These results are now included in **Supplementary Figure 3** and the corresponding section of the results. We also included a paragraph on mitochondrial dynamic changes in the axon in the discussion section.

Figure 1 to reviewers: examples of neurons expressing MitoTimer (pseudocolored as a red/green ratio) and cytosolic BFP as a marker of neuronal electroporation. In (A) MitoTimer was expressed in neurons before induction with doxycycline. In (B) there was no expression of MitoTimer despite the use of doxycycline.

2. Given the metabolic requirement of mitochondria for branching that the authors characterized so well, it was curious to me why the authors did not go further with their studies of BRAWNIN over- and under-expression. Does inhibiting BRAWNIN impair mitochondrial respiration similar to NUAK1 KO (Fig. 3)? Does reintroducing BRAWNIN in NUAK1 KO cells maintain mitochondrial respiration?

Response:

We agree this experiment was the logical next step to our study. However, it required a certain degree of technical optimization. Indeed, measurement of mitochondrial respiration with the Seahorse analyzer required to achieve knockdown in all the cells in the dish, which we implemented by optimizing the use of lentiviruses with an shRNA against BRAWNIN. The resulting Seahorse analysis (**Figure 8, panels F-H**) confirms a major decrease in mitochondrial respiration in cortical neurons upon inhibition of *Brawnin* expression. Of note the degree to which we knockdown *Brawnin* using the shRNA strategy is stronger than the changes observed in *Nuak1* cKO neurons.

Using again a lentiviral strategy, we subsequently attempted to rescue mitochondrial respiration upon re-expression of BRAWNIN. To this end, we generated a novel mouse BRAWNIN vector, and validated both viral production and BRAWNIN overexpression at mRNA level. Importantly, the strong CAG promoter which we used previously (**Figure 8, panels I-O**) was considered not suitable for lentivirus-based experiment, as previous reports describe a low encapsidation rate in CAG-containing plasmids. Thus, we cloned mouse BRAWNIN in a vector with an alternative promoter, which we never tested before. Unfortunately, we did not observe an increase in neuronal respiration upon overexpression of BRAWNIN in wild-type neurons. However, attempts to measure BRAWNIN overexpression at the protein level were unsuccessful, as commercial antibodies do not cross-react with mouse BRAWNIN.

We therefore think this rescue experiment will require more troubleshooting to optimize the use of BRAWNIN-expressing viruses, find the proper promoter and viral titer to drive a significant protein expression of BRAWNIN in primary neurons in a Seahorse plate. Furthermore, NUAK1 cKO neuronal cultures were a precious resource that depended upon breeding and genotypes in our mouse facility. As a consequence, we decided we could not move forward in the timeframe of the revisions for this paper.

However, we have performed respiration measurements upon rescue with L-Car as was suggested by Reviewer 3, and this data is now presented in **Supplementary Figure 7**. We think these experiments further strengthen the link between NUAK1, the regulation of mitochondrial

metabolic activity, and axonal branches stabilization.

Minor

- P8, Lines 13, 15: lower case “l” is used in the text in contrast to the upper case “L” in the figure.
- Based on the text on P12, L14-6, are there no statistical significance to report in Fig. 6 I and J?

Response:

The typos have been corrected. Statistical significance has been added to the indicated panels (now Fig 7E and F).

Reviewer #2 (Remarks to the Author):

This study offers a powerful argument that the Nauk1 kinase is fundamental in safeguarding appropriate mitochondrial metabolism in branching axonal collaterals of cortical neurons. It is strongly suggested that this function is carried out, at least in part, through the expression and splicing of the BRAWNIN protein in maintaining appropriate mitochondrial functions during the morphogenesis of cortical neurons.

The nuanced dance of activation of mitochondrial metabolism, affecting cell morphogenesis and cell biology, is a known phenomenon in this and related systems. However, what sets this research apart is the outstanding caliber of work, the significant relevance of neurodevelopmental disorders, and the exciting potential for these findings to be applied to other brain areas. In particular, the carefully designed experiments, the interweaving of in vivo data, and the well-executed epistasis experiments were very appealing. In the realm of neural development, this study is a noteworthy and praiseworthy addition.

In the face of numerous experiments that naturally invites the possibility to level a degree of critique, this reviewer chooses to refrain from nitpicking. The rationale behind this decision is the steadfast belief that minor critiques would hardly alter the robust core message presented in the paper. Rather than frittering away resources on minor adjustments, it would be far more beneficial for the research team to channel these toward their forthcoming studies.

Response:

We thank reviewer #2 for this very positive comment and the overall appreciation of our work. We have continued working on this project and provide additional data which strengthen the manuscript, and we hope this reviewer will appreciate the new results, which he can discover in our responses to reviewers #2&3.

Reviewer #3 (Remarks to the Author):

In this manuscript, the authors study the role of mitochondrial trafficking in axon branching, focusing on the role of NUAK1, a kinase they previously found to be involved in these processes. They first observe that mitochondrial stalling in axons is associated with stabilisation of new branches as the mitochondria enter the branches, rather than with the initial branch formation. Next, they study differential requirements for AMPK and NUAK1, and find no requirement for AMPK in their measures of mitochondrial stalling and axon branching, in vivo and in primary neuron cultures. In parallel, they confirm previous observations that AICAR, a widely used AMPK activator, is not specific for AMPK, but also activates the AMPK-like NUAK1. Finally, based on RNA-seq from WT and NUAK1-mutant cortices and neuronal cultures, they identify Brawnin, a mitochondrial localised peptide, as a key mediator of NUAK1-induced mitochondrial activity in axons, and subsequent branching phenotypes. This is a solid manuscript, with really interesting and novel findings, providing significant new insight into how mitochondrial metabolism and trafficking interact with axon branching and how this is regulated. The manuscript is well written, with clear and convincing figures, and the experiments seem overall well conducted, with different types of experiments agreeing and supporting the conclusions made throughout.

I am supportive of publication, but have several concerns, mostly minor, listed below, that should be addressed first.

1. Some of the conclusions are overstated, or imply causality when the experiments merely show a correlation. Examples:

- Page 5 line 24: 'branches that failed to recruit rapidly retracted', while the experiments show a correlation in time.
- Page 8 line 19-21: 'a function that can be uncoupled from its role in trafficking'. It's not clear to me how this can be stated, without rescuing mitochondrial trafficking in NUAK1 mutant neurons. This experiment is not required for publication, but the interpretation is not correct.
- Page 12 line 17-20: the experiments do not address direct impact of NUAK1 on Brawnin splicing, and differential splicing could have many alternative explanations.

Response:

We thank reviewer #3 for the critical reading of our manuscript. We reformulated the parts according to this reviewer's comment.

2. Fig. 1O,P: comparison and statistical analysis should be done on +/- mitoKillerRed as well (i.e. comparing O vs P, ideally in the same graph).

Response:

In Figure 1O-P, we measured the percentage of branches elongating (in green) per axon. Thus, in this experiment, each axon is its own control, which is why we compared branch dynamics inside and outside of the ROI. As requested, we reanalyzed the data to compare branch dynamics outside of the ROI in neurons electroporated with mito-KillerRed (O) and mito-DsRed (P) and saw no difference, which further demonstrates that mito-KillerRed has no toxicity and no impact on axon branching unless it is photoactivated. This quantification is presented in **Supplementary Figure 2J**.

3. Fig. 3A-D: How come the mtDNA content is not different, if the axonal length and branch number are so much lower in the NUAK1 KO neurons? Where have the mitochondria gone then? Is the soma larger, or are there mitochondria per cell volume?

Response:

We agree with the observation from this reviewer and don't have a definite explanation at this stage. As suggested, we measured soma surface area, somatic mitochondria occupancy (surface area occupied by mitochondria, which was a more robust measurement than mitochondria number given that somatic mitochondria are fused), as well as the density of axonal mitochondria, and we compared WT and KO neurons. Results are presented in **Supplementary Figure 3 (panels F-H)**. We didn't detect any increase of somatic mitochondria content in the KO condition that would compensate for the loss of axonal mitochondria. Furthermore, and as discussed above, we didn't detect changes in mitochondria biogenesis or turnover.

Of note, there were on average 4.9 mitochondria per 100µm of axon length in the control condition. In a typical experiment run in the lab, we measure on average axon length at 300µm in control condition, which is 450µm of total length including collaterals. A 50% reduction in axon length would represent the loss of roughly $(4.9 \times 4.5 / 2 =)$ 11 mitochondria, which would be extremely hard to detect in the soma and/or on mtDNA content, given the cell-cell variability.

4. Fig. 3I: What is the source of the increased ROS production if the OxPhos activity is significantly decreased? The fig references on page 8 are not always correct (eg I instead of M).

Response:

Previous studies demonstrated that the assembly of the mitochondrial respiratory chain determines the efficiency of the OxPhos and ROS production. Especially, differences into the assembly of supercomplexes can explain lower mitochondrial respiration and higher ROS production in astrocytes compared to neurons⁶⁻⁸. This observation could explain the coincidental occurrence of lower respiratory capacity of *Nuak1* KO neurons and higher levels of ROS. We have not checked if the assembly of the supercomplexes is altered in *Nuak1* KO neurons, which falls out of the scope of the present study and will be the focus of future investigations, but we cannot rule out this possibility especially since BRAWNIN has been involved in the mitochondrial respiratory chain assembly.

5. Fig. 3H-K: I could not find how the Perceval signal was quantified and normalised. Values from these genetically encoded sensors can be highly variable and sometimes difficult to interpret. The authors should show the raw (non-normalised) values in a supplementary figure, and explain better what is

meant by the F0/Fmin values in Fig. 3.

Response:

As requested, we provide an explanation of the quantification in the supplementary methods section, and we provide the raw data as **Supplementary Figure 11**.

6. Fig. 4F-H: the authors should address (or discuss) the possibility that AICAR may act on LKB1, upstream to NUA1, rather than on NUA1 directly.

Response:

AICAR has long been considered a relatively specific activator of AMPK. Work by the group of Dario Alessi demonstrated in rat muscles that AICAR does not activate LKB1 (nor selected AMPK-related kinases – importantly, the paper does not measure NUA1 response to AICAR)⁹. Furthermore, it has been reported that AICAR effect on AMPK can be LKB1 independent in cell lines that don't express it such as HeLa cells¹⁰. For these reasons a direct activation of LKB1 by AICAR is not the most likely mechanism. We discuss this point in the manuscript.

7. Page 9-10 and Fig. 5 are interpreted too optimistic.

- The statement that L-Car is a 'mitochondria modulator', and upregulates mitochondrial function by promoting LCFA metabolism is an over-simplification. Axonal OxPhos activity could well be driven by (lactate-derived?) pyruvate and TCA-cycle activity, rather than FA oxidation.

- It is not clear how increasing FA metabolism by LCAR (if this is indeed what happens) can increase OxPhos activity if Fig. 3 already demonstrates decreased maximal respiration?

(...)

The authors should either restrict their interpretations to description of the experiments, or should perform additional experiments: (1) measure mitochondrial OxPhos activity and PDH phosphorylation upon LCAR and MK 4 treatment; (2) overexpress or inactivate LDHA vs B to activate lactate/pyruvate-driven TCA-cycle activity; (...).

Response:

As suggested, we performed Seahorse analyses in neurons treated with L-Car. These results are now presented as **Supplementary Figure 7A-C**. They show that L-Car increases the basal (although not significantly) and maximal ($p < 0.05$) respiration in NUA1 cKO neurons, up to the level of control neurons.

Next, we sought to determine how L-Car acts on mitochondrial respiration. We used the long chain fatty acid oxidization stress kit from Seahorse, which relies on the inhibition of CPT-1 by the drug Etomoxir, to measure how much developing cortical neurons rely on FA oxidization for mitochondrial respiration. Our results confirmed that cortical neurons use very little fatty acids for their mitochondrial respiration (not shown), which was expected from the literature. On the contrary, the blockage of pyruvate import into the mitochondria with UK-5099 (Glucose/pyruvate oxidization stress kit) led to a more pronounced decrease in mitochondrial respiration. These results are in line with the general consensus that neurons rely essentially on glucose/pyruvate metabolism for their energetic need. However, when neurons were treated with L-Car, we could observe an increase in mitochondrial respiration (basal and maximal) even in the presence of UK-5099. These results, presented in **Supplementary Figure 7D-F**, do not establish that L-Car acts by promoting FA oxidization. However, they suggest that the activity of L-Car does not rely solely on pyruvate catabolism to fuel the TCA.

Thus, we think our data support a pro-metabolic effect of L-Car, which cannot be explained by the preferential use of pyruvate by neuronal mitochondria. However, we agree with the suggestion that L-Car may play several roles and that our work does not fully address the mechanism by which L-Car regulates neuronal mitochondrial functions, directly or indirectly. Thus, we toned down our interpretation to be more cautious about the effects of L-Car.

- I am not convinced that the quantification of contralateral axonal density after IUE is an appropriate measure of axon branching. This is also the case for Fig. 7F-M.

The authors should (...) (3) quantify axonal density in the corpus callosum as a measure of axon length, and explain how variability between different IUE experiments is accounted for.

Response:

We now provide a better description of the quantification method following *in utero* cortical

electroporation in the methods section. As mentioned by this reviewer, the variability from one electroporation to the other is a challenge in quantifying cortical axons from brain slices. In all our studies, we normalize the signal (fluorescence intensity, as a measure of the density of axons in the ipsilateral or contralateral cortex) by the fluorescence intensity in the corpus callosum, which allows to correct for variations in the number of electroporated brains from one animal to the next. This normalization method also corrects for changes in signal intensity from one experiment to the other which can be caused by changes in tissue preparation (which is highly dependent upon intracardiac perfusion and thus can vary from one animal to the other), antibody variability or microscope settings.

We tested that this normalization method is efficient by measuring, on slices prepared from control animals, the fluorescence intensity (GFP) in the corpus callosum, and observed that it is directly correlated to the number of electroporated cells (counted on the same brain slice) (see **Figure 2 to reviewers, panel A**). Hence, this confirms that data normalization by fluorescence intensity in the corpus callosum is a valid method to correct for variation in transfection efficiency.

As requested, we also measured axonal density in the corpus callosum in **Figure 5F-I** and in **Figure 7G-L**. These quantifications are presented in **Figure 2 to reviewers, panels B and C**. Our interpretation of this data is that they reveal differences in electroporation efficiency, staining and microscope settings rather than an effect on axon growth. More importantly, this data does not change the interpretation from axon branching (ipsilateral and contralateral) since normalization blunts sample to sample difference. Hence, we decided not to include the quantification of fluorescence intensity in the corpus callosum in the article, and rather present these data in the **Figure 2 to reviewers** below.

Figure 2 to reviewers: (A) correlation between GFP intensity in the corpus callosum (arbitrary units) and the number of electroporated neurons found on the same brain slice. (B) quantification of the signal intensity of *mscarlet-i* in the corpus callosum for animals presented in Figure 5F-I. Although there was a decrease of signal in the KO condition compared to the control (WT), note that this was the case also for animals treated with L-Car (both WT and KO) despite having a high density of terminal axon branches. (C) quantification of GFP signal intensity in the corpus callosum for animals presented in Figure 7G-L. Data: average + 95%CI, Statistical tests: one-way ANOVA with Bonferroni's multiple comparison.

8. Fig. 7R,S,T: NUAK1KO/sh + Brawnin should be compared to NUAK1KO/sh alone as well.

Response:

We performed the tests as requested.

Additional references

1. Vaarmann, A. *et al.* Mitochondrial biogenesis is required for axonal growth. *Development (Cambridge, England)* **143**, 1981–1992 (2016).

2. Lewis, T. L., Kwon, S.-K., Lee, A., Shaw, R. & Polleux, F. MFF-dependent mitochondrial fission regulates presynaptic release and axon branching by limiting axonal mitochondria size. *Nature communications* **9**, 5008 (2018).
3. Cheng, X.-T., Huang, N. & Sheng, Z.-H. Programming axonal mitochondrial maintenance and bioenergetics in neurodegeneration and regeneration. *Neuron* **110**, 1899–1923 (2022).
4. Brooks, D. *et al.* Drosophila NUAK functions with Starvin/BAG3 in autophagic protein turnover. *PLOS Genetics* **16**, e1008700 (2020).
5. Hernandez, G. *et al.* MitoTimer. *Autophagy* **9**, 1852–1861 (2013).
6. Lopez-Fabuel, I. *et al.* Complex I assembly into supercomplexes determines differential mitochondrial ROS production in neurons and astrocytes. *Proc. Natl. Acad. Sci.* **113**, 13063–13068 (2016).
7. Morant-Ferrando, B. *et al.* Fatty acid oxidation organizes mitochondrial supercomplexes to sustain astrocytic ROS and cognition. *Nat. Metab.* **5**, 1290–1302 (2023).
8. Vicente-Gutierrez, C. *et al.* Astrocytic mitochondrial ROS modulate brain metabolism and mouse behaviour. *Nature Metabolism* **1**, 201–211 (2019).
9. Sakamoto, K., Göransson, O., Hardie, D. G. & Alessi, D. R. Activity of LKB1 and AMPK-related kinases in skeletal muscle: effects of contraction, phenformin, and AICAR. *Am. J. Physiol.-Endocrinol. Metab.* **287**, E310–E317 (2004).
10. Sun, Y., Connors, K. E. & Yang, D.-Q. AICAR induces phosphorylation of AMPK in an ATM-dependent, LKB1-independent manner. *Mol. Cell. Biochem.* **306**, 239–245 (2007).

REVIEWERS' COMMENTS

Reviewer #1 (Remarks to the Author):

I appreciate the authors' efforts to experimentally address my previous critiques. The manuscript is stronger for their efforts. Well done.

Reviewer #3 (Remarks to the Author):

The authors responded thoroughly to all my concerns, and I am supportive for publication.